# The SPACE 1.0 model: A Landlab component for 2-D calculation of sediment transport, bedrock erosion, and landscape evolution

Charles M. Shobe[1], Gregory E. Tucker[1], and Katherine R. Barnhart[1]

[1]CIRES and Department of Geological Sciences, University of Colorado

*Correspondence to:* Charles M. Shobe (charles.shobe@colorado.edu)

**Abstract.** Models of landscape evolution by river erosion are often either transport-limited (sediment is always available, but may or may not be transportable) or detachment-limited (sediment must be detached from the bed, but is then always transportable). While several models incorporate elements of, or transition between, transport-limited and detachment-limited behavior, most require that either sediment or bedrock, but not both, are eroded at any given time. Modeling landscape evolution over large spatial and temporal scales requires a model that can 1) transition freely between transport-limited and detachment-limited behavior, 2) simultaneously treat sediment transport and bedrock erosion, and 3) run in 2-D over large grids and be coupled with other surface process models. We present SPACE (Stream Power with Alluvium Conservation and Entrainment) 1.0, a new model for simultaneous evolution of an alluvium layer and a bedrock bed based on conservation of sediment mass both on the bed and in the water column. The model treats sediment transport and bedrock erosion simultaneously, embracing the reality that many rivers (even those commonly defined as "bedrock" rivers) flow over a partially alluviated bed. SPACE improves on previous models of bedrock-alluvial rivers by explicitly calculating sediment erosion and deposition rather than relying on a flux-divergence (Exner) approach. The SPACE model is a component of the Landlab modeling toolkit, a Python-language library used to create models of Earth surface processes. Landlab allows efficient coupling between the SPACE model and components simulating basin hydrology, hillslope evolution, weathering, lithospheric flexure, and other surface processes. Here, we first derive the governing equations of the SPACE model from existing sediment transport and bedrock erosion formulations and explore the behavior of local analytical solutions for sediment flux and alluvium thickness. We derive steady-state analytical solutions for channel slope, alluvium thickness, and sediment flux, and show that SPACE matches predicted behavior in detachment-limited, transport-limited, and mixed conditions. We provide an example of landscape evolution modeling in which SPACE is coupled with hillslope diffusion, and demonstrate that SPACE provides an effective framework for simultaneously modeling 2-D sediment transport and bedrock erosion.

## 1 Introduction

Rivers are the primary agents of land-surface lowering in non-glaciated landscapes (e.g., Whipple, 2004). Erosion and sediment transport in rivers affect human river management (e.g., Graf et al., 2010), landscape mass balance (e.g., Armitage et al., 2011), and global biogeochemical cycling (e.g., Hilton, 2017). Interest in the effects of river erosion on landscape change over all spatial and temporal scales has led to the widespread proliferation of numerical models for channel evolution. Specifically

within the landscape evolution community, many models have emerged to address the problem of river incision into sediment and bedrock over long timescales. River incision has commonly been modeled with one of two end-members: transport-limited and detachment-limited models. Transport-limited models (e.g., Willgoose et al., 1991) assume that erosion is limited by the capacity of a river to transport sediment or rock, but that an infinite supply of readily transportable material is available. Transport-limited models do not explicitly incorporate a formulation for the mechanics of bed erosion, but assume that elevation change in the channel is set by the divergence of sediment transport capacity (e.g., Paola and Voller, 2005). Detachment-limited models (e.g., Howard, 1994) assume that erosion is limited by a river's ability to remove sediment or rock from the bed, but that all detached material is transportable. As such, they include no statement of mass conservation (aside from the assumption that all eroded mass leaves the model domain) because all detached mass is assumed to be transported downstream. Transport capacity in transport-limited models and detachment rate in detachment-limited models are generally assumed to be some function of water quantity and slope (e.g., shear stress or stream power).

In this paper, we briefly discuss the problem of differentiating between these two end-member models, and how previous workers have attempted model validation in the field. We review existing models that fall between the two end-members, and argue that many models are limited by an inability to treat simultaneous sediment transport and bedrock erosion over landscape evolution timescales. We then describe the SPACE 1.0 model, a new channel evolution model intended to simulate the long-term evolution of bedrock-alluvial channels at the landscape scale. SPACE calculates sediment transport and bedrock erosion simultaneously by modeling the coupled evolution of an alluvial layer and a bedrock bed. We show that the SPACE model transitions smoothly between transport-limited, detachment-limited, and mixed behavior, and that it matches analytical solutions for channel slope, sediment layer thickness, and sediment flux. We use the example of topographic evolution in response to unsteady rock uplift to show that as a component of the Landlab toolkit, SPACE may be coupled with other landscape evolution model components to explore landscape change over long timescales. Finally, we discuss the potential for field validation of the SPACE model, the limitations of the model, and the similarities and differences between SPACE and previous channel evolution models.

## 2    The problem of model differentiation and applicability

While the underlying assumptions of the transport-limited and detachment-limited end-member models differ, they produce identical steady-state longitudinal profiles (e.g., Whipple and Tucker, 2002; Lague et al., 2003). Further, as noted by Davy and Lague (2009), the applicability of one end-member model or the other to a given landscape appears to depend on the characteristics of test sites and on the comparison methods (e.g., Snyder et al., 2003a; Tomkin et al., 2003; van der Beek and Bishop, 2003; Valla et al., 2010; Hobley et al., 2011). Such tests generally consist of using models to match topographic variables such as erosion rate–channel steepness scaling (e.g., Snyder et al., 2003a), matching steady-state model solutions to river profiles with evidence for steady-state behavior (e.g., Tomkin et al., 2003), using a well-constrained initial condition and present-day topography to compare river profile evolution between models and the landscape (e.g., van der Beek and Bishop, 2003; Valla et al., 2010), or combining fluvial erosion depths with field model calibration to find an optimal model form (e.g.,

Hobley et al., 2011). Both end-member models have been shown to be applicable in certain landscapes. For example, both Valla et al. (2010) and Hobley et al. (2011) investigated the carving of postglacial gorges (in the Alps and the Himalaya, respectively) and both used gorge incision depth and other field measurements to validate transport-limited and detachment-limited incision models. Valla et al. (2010) concluded that a transport-limited model was most appropriate for treating gorge incision at their study site, while Hobley et al. (2011) found that gorge evolution in the Himalaya was best replicated with a detachment-limited model. Neither the detachment-limited nor the transport-limited end-member model has been shown to agree with field data in a wide variety of natural settings, likely due in large part to the potential for both types of behavior to occur simultaneously. This suggests that to apply to real landscapes, landscape evolution models must include both sediment mass conservation (as in transport-limited models) and sediment production/bed erosion (as in detachment-limited models). Below we review some of these models, specifically focusing on those that treat both sediment transport and bedrock erosion.

## 3 Review of sediment transport and bedrock erosion models

Over the past two decades, new efforts to advance modeling of bedrock-alluvial river channel evolution have focused on 1) improving descriptions of sediment mass conservation and transport, and 2) improving formulations for bed erosion. Additionally, several models have been developed that combine statements of mass conservation and bed erosion in order to simulate simultaneous sediment transport and bed evolution. We focus specifically on those models combining sediment dynamics and erosion into bedrock, as our goal is the development of a model for the evolution of mixed bedrock-alluvial channels. Substantial advances have also been made in modeling the hydrodynamics that drive channel evolution, but we do not review that work here.

### 3.1 Approaches to mass conservation

Realistic treatments of sediment morphodynamics are critical to models of bedrock-alluvial river evolution. Incorporating conservation of sediment mass leads to process-based descriptions of sediment flux and sediment cover on bedrock channel beds. The two approaches to sediment mass conservation considered here are the Exner-type approach, in which temporal changes in sediment thickness are driven by changes in sediment transport capacity, and the erosion-deposition approach, in which erosion and deposition fluxes of sediment are calculated to determine changes in sediment flux and alluvial layer thickness. Either type of mass conservation statement may be used to model bedrock-alluvial rivers. Inoue et al. (2014) used a modified 1-D Exner equation for sediment mass conservation incorporating both mobile bedload and an alluvial layer. Lague (2010) and Zhang et al. (2015) followed a similar approach, further modifying the sediment mass conservation framework to include below-capacity sediment transport. Exner sediment transport in bedrock-alluvial systems was expanded to 2-D by Nelson and Seminara (2012), Inoue et al. (2016), and Inoue et al. (2017), though the specifics of calculating sediment flux differ between the model of Nelson and Seminara (2012) and the latter two. The Exner-based conservation framework allows calculation of bed cover by sediment as a function of the ratio of sediment flux to transport capacity $q_s/q_c$ (Inoue et al., 2014). Inoue et al. (2014) predicted a range of cover functions depending on the ratio of grain roughness to bedrock roughness,

ranging from an exponential decline in exposed bed fraction with increasing $q_s/q_c$ to rapid alluviation above a $q_s/q_c$ threshold. Other models for alluvial bed cover arise from simplifications to the Exner-style approach. Johnson (2014) made steady-state predictions for alluvial cover as a function of $q_s/q_c$ on a non-erodible bedrock bed, and showed that bedrock roughness relative to sediment size is an important control on setting the influence of $q_s/q_c$ on bed cover. Nelson and Seminara (2011) assumed that sediment was transported at capacity at every point along a 2-D channel cross-section in order to compute the thickness of alluvial cover in the cross-section.

The erosion-deposition framework differs from the Exner approach in that it conserves mass on the bed and in the water column to treat simultaneous erosion and deposition of a single substrate (Beaumont et al., 1992; Braun and Sambridge, 1997; Coulthard et al., 2002; Davy and Lague, 2009; Carretier et al., 2016). Erosion-deposition mass conservation is based on explicit calculations of sediment entrainment and deposition, rather than on the spatial divergence of sediment transport capacity. Erosion-deposition models may dynamically transition between transport-limited and detachment-limited behavior (see discussion by Davy and Lague (2009)). Davy and Lague (2009) present such a model based on the relative influence of erosion from the bed into the water column (erosion flux) and deposition from the water column onto the bed (deposition flux). When the deposition flux is much smaller than the erosion flux and the sediment transport length scale, which can be thought of as the average travel distance of a grain from entrainment to re-deposition, is long, their model becomes equivalent to a basic detachment-limited model. When the deposition flux increases and the transport length scale is short, the model predicts transport-limited behavior in which sediment flux divergence controls the channel bed elevation. The erosion-deposition framework, validated for alluvial rivers by the laboratory experiments of Lajeunesse et al. (2017), is capable of matching transient and steady-state longitudinal profile predictions made by both detachment-limited and transport-limited models, and smoothly transitions between the two types of model behavior (Davy and Lague, 2009). The model of Davy and Lague (2009) improved on previous erosion-deposition models (Beaumont et al., 1992) by using a sediment transport length scale that increases with water discharge. Erosion-deposition-type models that transition between detachment-limited and transport-limited behavior allow the exploration of both types of models, and intermediate cases, with simple parameter changes rather than changes to the model structure.

Erosion-deposition models, like Exner-based transport models, can yield predictions for the influence of sediment dynamics on erosion-inhibiting sediment bed cover. The models of Hodge and Hoey (2012), Nelson and Seminara (2012), and Turowski and Hodge (2017) incorporate statements of sediment mass conservation to explore cover dynamics on a stationary bedrock bed. Hodge and Hoey (2012) used a cellular automaton model of alluvial cover evolution that allowed probabilistic entrainment and deposition of sediment on a non-eroding bedrock bed. They found that either a linear or exponential decline in bedrock exposure with increasing $q_s/q_c$ could occur if all grains had equal entrainment probability, with the exponential decline occurring at low entrainment probabilities and the linear decline occurring at high entrainment probabilities. Additionally, they found that when isolated grains have a higher entrainment probability than clustered grains, bedrock exposure declines sigmoidally with increasing $q_s/q_c$ and runaway alluviation can occur. Their results support the experimental results of Chatanantavet and Parker (2008) showing that both the linear decline in exposure with increasing $q_s/q_c$ and the abrupt shift from fully exposed to fully alluviated bed are possible, with the former occurring at low slopes and high bed roughness, and the latter occurring at high

slopes and low bed roughness. Turowski and Hodge (2017) explicitly tracked erosion and deposition governing the transfer of particles between two mass reservoirs: the stationary particles on a non-erodible bedrock bed and the mobile particles in the water column. They showed that the relationship between bedrock exposure and $q_s/q_c$ can take a wide range of forms depending on, among other factors, the probability of increasing bed cover for a given fraction of exposed bed.

## 3.2 Approaches to bedrock erosion

Evolving the channel bed over time requires a description of the mechanics of bed erosion, which can be coupled with sediment mass conservation to model bedrock-alluvial channels. The most well-known approach to bed erosion is the stream power and shear stress family of models (e.g., Howard, 1994; Whipple and Tucker, 1999), in which bed erosion is a function of water quantity (either depth or discharge) and channel bed slope. The major limitation of basic stream power and shear stress models is the lack of a description of how sediment influences channel bed erosion. A large number of sediment-flux-dependent incision models, in which the interaction of sediment flux with sediment transport capacity enhances or inhibits bedrock erosion, have been developed to fill this gap (e.g., Sklar and Dietrich, 1998; Whipple and Tucker, 2002; Sklar and Dietrich, 2004; Gasparini et al., 2006, 2007; Turowski et al., 2007; Chatanantavet and Parker, 2009; Hobley et al., 2011). Sediment may act as "cover," inhibiting bedrock erosion, or as "tools," accelerating bedrock erosion (see review by Hobley et al. (2011)). In most sediment-flux dependent models, erosion is influenced by a factor $f(q_s, q_c)$ ranging between 0 and 1, where $q_s$ is sediment flux (either volume or mass flux per unit width) and $q_c$ is sediment transport capacity. The value of $f(q_s, q_c)$ for any particular $q_s$ and $q_c$ depends on the choice of function $f$; proposed forms include a linear decline ($f(q_s, q_c) = 1 - \frac{q_s}{q_c}$; Beaumont et al. (1992)), parabola (Sklar and Dietrich, 2004), near-parabola (Gasparini et al., 2006), and other similar shapes (Turowski et al., 2007). Turowski et al. (2007) showed that the fraction of exposed bed may decline exponentially with increasing $q_s/q_c$, leading to an exponential tail on $f(q_s, q_c)$. Any of the formulations for bedrock erosion detailed here are insufficient on their own for modeling bedrock-alluvial channels, as there is no mass conservation of sediment and therefore no development of an alluvial layer. Any bedrock erosion model may be coupled with a statement of mass conservation to capture the coupled evolution of bedrock-alluvial channels.

## 3.3 Coupled sediment transport and bedrock erosion models

While several existing erosion-deposition models contain descriptions of both sediment mass conservation and bed erosion, some are limited to erosion through a single substrate, and are not capable of evolving a bedrock-alluvial channel with a distinct alluvial layer (e.g., Davy and Lague, 2009; Carretier et al., 2016). Carretier et al. (2016) presented a model in which the erosion-deposition framework is applied to cases in which a layer of sediment overlies bedrock, but their model requires that one or the other material fully occupy the surface of a cell at a given time. Without substantial modification, erosion-deposition models are therefore unable to simultaneously treat sediment morphodynamics and bedrock erosion in mixed bedrock-alluvial systems. A small subset of existing channel evolution models combine sediment mass conservation with equations for bedrock incision to treat bedrock-alluvial channels. For example, Lague (2010) used the Exner equation for sediment conservation to explicitly track a layer of alluvium of thickness $T_s$ with median grain size $D_{50}$ overlying a bedrock bed. He tested both the linear and the

exponential form of the decline in bed exposure with increasing $q_s/q_c$, both expressed as a function of the ratio of $T_s$ to $D_{50}$ (his Fig. 3). The model of Lague (2010) partitions stresses between the channel bed and banks (Flintham and Carling, 1988) to allow dynamic channel width variations, but does not incorporate the tools effect of sediment on bedrock erosion. Fowler et al. (2007), following in the footsteps of Smith and Bretherton (1972), combined the St. Venant equations with an Exner equation

for sediment mass conservation. Fowler et al. (2007) incorporated an equation for bed abrasion by bedload, assuming that abrasion rates should scale with the velocity of the bedload layer and should decline with increasing bedload layer thickness (yielding a form of the tools and cover effects). Several workers have adapted forms of the saltation-abrasion bedrock erosion model of Sklar and Dietrich (1998, 2004) in conjunction with sediment mass conservation to incorporate both the tools and cover effects into models of bedrock-alluvial river erosion (Turowski, 2009; Nelson and Seminara, 2011; Inoue et al., 2014,

2016, 2017). Turowski (2009) combined a stochastic erosion-deposition model with a saltation-abrasion-style incision rule and explored the sensitivity of bed cover and bedrock erosion rate to changes in sediment transport. Turowski (2009) showed an exponential decline in bed exposure with increasing numbers of sediment particles equivalent to the exponential decline in the deterministic model of Turowski et al. (2007). Nelson and Seminara (2011) coupled the cross-section channel flow model of Kean and Smith (2004) with a simple parameterization for the formation and destruction of an alluvial layer on the channel

bed. The thickness and lateral extent of the alluvial layer vary in response to the ratio of sediment flux to sediment transport capacity. Unlike most models described here, Nelson and Seminara (2011) explicitly resolved the shear stresses on the channel bed and banks and applied the saltation-abrasion model of bedrock erosion to exposed portions of the channel margin. Their model allows dynamic channel width, but is only solved at a single channel cross-section with a prescribed bed slope. This at-a-station approach precludes the development of alluvial cover by sediment derived from bedrock erosion. Inoue et al. (2014)

presented a 1-D model of the co-evolution of an alluvial layer and a bedrock bed. They used a modified Exner equation for sediment mass conservation incorporating both mobile bedload and an alluvial layer, and a saltation-abrasion-style bedrock incision rule (Sklar and Dietrich, 2004; Chatanantavet and Parker, 2009). Inoue et al. (2016, 2017) expanded the sediment conservation and bedrock erosion formulations of Inoue et al. (2014) to 2-D to allow spatial variability in alluvial thickness and bedrock erosion along both the channel length and width. Inoue et al. (2017) advanced the model of Inoue et al. (2016) by

incorporating a simple parameterization for bedrock bank erosion to investigate meander bend migration. Zhang et al. (2015) presented a channel longitudinal profile model coupling local sediment transport dynamics with the saltation-abrasion incision rule of Sklar and Dietrich (2004). The model of Zhang et al. (2015) can capture transience driven by changes in $q_s/q_c$, or downstream propagation of changes in sediment supply. Their model tracks the thickness of a sediment layer compared to the macro-roughness of the bedrock surface such that alluvial thickness less than the macro-roughness length-scale results in

exposed bedrock. Their geometric approach is similar to that of Lague (2010), but the Zhang et al. (2015) model incorporates both tools and cover effects due to its derivation from the saltation-abrasion model (Sklar and Dietrich, 2004). While Zhang et al. (2015) note that their model could incorporate downstream variations in channel width, it does not do so in the form they presented.

## 3.4 Reach-scale vs. landscape-scale approaches

The models reviewed above differ in their intended spatial and temporal scales of application. For example, the cross-section saltation-abrasion model of Nelson and Seminara (2011) resolves flow hydraulics and channel width variations with high fidelity, but cannot be feasibly applied at the landscape scale. Similarly, the bed cover evolution models of Turowski (2009), Hodge and Hoey (2012), Nelson and Seminara (2012), Johnson (2014), and Turowski and Hodge (2017) focus specifically on cover dynamics at the reach scale. While principles from these models could be incorporated into larger-scale landscape evolution models, several of the models (Hodge and Hoey, 2012; Nelson and Seminara, 2012; Johnson, 2014; Turowski and Hodge, 2017) are formulated for a fixed bedrock bed, thus limiting their potential for application to problems of long-term landscape evolution. The 2-D mixed bedrock-alluvial models proposed by Inoue et al. (2016, 2017) could potentially be incorporated into existing landscape evolution model frameworks, but the feasibility of such an integration is unclear as the models have been tested over hourly to daily timescales, not geologic timescales. The models most suited to landscape evolution applications are those of Fowler et al. (2007), Lague (2010), Inoue et al. (2014), and Zhang et al. (2015). The Inoue et al. (2014) model realistically incorporates hydraulic roughness of both bedrock and alluvium, but is designed to handle a constant sediment supply rate, a constraint which is unlikely to be satisfied in landscape evolution modeling applications. The model of Fowler et al. (2007) assumes no supply limitation on sediment flux, and is thus not suited to modeling mixed bedrock-alluvial cases. The two existing models that most closely match our goal of treating bedrock-alluvial channel dynamics over landscape evolution space and time scales are those of Lague (2010) and Zhang et al. (2015). We advance on these models by using the erosion-deposition sediment conservation framework to craft explicit expressions for the entrainment and deposition of sediment as well as the erosion of bedrock.

## 3.5 The SPACE model

We present a new model for simultaneous sediment and bedrock evolution that extends from the approaches of Davy and Lague (2009), Lague (2010), and Zhang et al. (2015), as well as from the foundation laid by the other models reviewed above. The Stream Power with Alluvium Conservation and Entrainment (SPACE) model conserves sediment in two reservoirs, the bed and the water column, in the style of Davy and Lague (2009). It is therefore an erosion-deposition model in its treatment of sediment. SPACE also calculates bedrock erosion, incorporating progressive bedrock exposure with thinning of an alluvial layer in a similar way to Lague (2010) and Zhang et al. (2015) such that evolution of the alluvium and the bedrock may be simultaneously calculated. SPACE is unique in that it employs local analytical solutions for sediment flux changes in space and alluvium thickness changes in time. These local analytical solutions are derived by combining expressions for conservation of sediment in the water column (e.g., Davy and Lague, 2009), conservation of sediment on the channel bed, and conservation of mass of bedrock. Our new model advances beyond sediment-flux-dependent bedrock incision models in which sediment transport and storage are not treated explicitly, as well as previous erosion-deposition models assumed to be eroding a single substrate. The SPACE model is designed to run over large spatial and temporal scales in two dimensions, and can be easily coupled to many other surface process models as part of the Landlab modeling toolkit.

To maintain ease of model coupling and a relatively low level of complexity, we neglect some processes treated in previous models. Because sediment cover and transport are averaged across a model cell, we do not explicitly model the spatial distribution of sediment cover within a cell, or the entrainment and deposition of individual sediment grains (e.g., Turowski, 2009; Hodge et al., 2011; Hodge and Hoey, 2012; Hodge, 2017). We do not employ dynamic channel width variations as in the models of Davy and Lague (2009), Lague (2010), Nelson and Seminara (2011), and Coulthard et al. (2013), instead relying on empirical parameterizations of channel width as a function of drainage area or discharge. SPACE does not contain independent descriptions of the roughness of bedrock and sediment, and does not distinguish between the case in which bedrock is rougher than sediment and the one in which sediment is rougher than bedrock (e.g., Inoue et al., 2014; Johnson, 2014). We also do not model the potential driving of bedrock erosion by bedload sediment (the tools effect), as used by Sklar and Dietrich (2004), Chatanantavet and Parker (2009), Turowski (2009), Inoue et al. (2014, 2016, 2017), Zhang et al. (2015), and other saltation-abrasion type models. Both width dynamics and bedload abrasion could be incorporated without changing the underlying model structure, but have been omitted in order to facilitate numerical model comparison to analytical solutions.

The SPACE model is intended to model river channel evolution over long timescales while honoring the reality that channels may transition between alluviated, bedrock, and mixed bedrock-alluvial states over geologic time. Below we develop the SPACE model, verify our numerical solutions against analytical solutions, and show that SPACE can transition naturally between transport-limited and detachment-limited behavior. We briefly describe the Landlab modeling toolkit, and show an example landscape evolution model in which SPACE is coupled to other surface process models.

## 4    SPACE model development

The SPACE model, like other erosion-deposition models, arises from sediment mass conservation in the water column (e.g., Davy and Lague, 2009) and on the channel bed. We consider a river bed that may vary dynamically in its relative proportion of alluvial cover and exposed bedrock, with a bedrock surface of height $R$, bed sediment of mean thickness $H$ (the channel bed elevation $\eta$ is therefore $R + H$), and water of mean flow depth $h$ (Fig. 1). We use the term "SPACE model" to refer to the model equations, and "SPACE 1.0 component" to refer specifically to the numerical implementation of the SPACE model as a Landlab component.

### 4.1    Conservation of sediment in the water column

Davy and Lague (2009) showed that the rate of change in the volume of sediment in the water column per unit area of river bed $\frac{\partial (c_s h)}{\partial t}$ may be written as:

$$\frac{\partial (c_s h)}{\partial t} = E_s - D_s - \frac{\partial (Q_s/w)}{\partial x}, \tag{1}$$

where $c_s$ is the concentration of sediment in the water column calculated by $Q_s/Q$ (where $Q$ is volumetric water discharge), $E_s$ is the volumetric entrainment flux of sediment per unit bed area, $D_s$ is the volumetric deposition flux of sediment per unit bed area, $Q_s$ is sediment flux in units of $L^3/T$, and $w$ is channel width. Eq. (1) is sufficient when considering a channel bed

on which only a single material (i.e., sediment) is exposed. Yet many channels are partially alluviated, indicating that sediment and bedrock may be eroded and entrained into the water column simultaneously. Such a scenario requires an addition to Eq. (1) to account for entrainment of bedrock material into the water column:

$$\frac{\partial \left(c_s h\right)}{\partial t} = E_s + \left(1 - F_f\right) E_r - D_s - \frac{\partial \left(Q_s/w\right)}{\partial x}. \tag{2}$$

$E_r$ is the volumetric erosion flux of bedrock per unit bed area. $F_f$ is a unitless fraction of fine sediment. $F_f$ represents the volumetric fraction of bedrock that breaks into sediment small enough to be considered permanently in suspension, for which no further treatment of bed–water column interactions is needed. For bedrock that breaks only into sand and gravel fractions, $F_f$ would be zero. Therefore, bed sediment thickness $H$ and sediment flux $Q_s$ only include sediment coarse enough that it does not enter permanent suspension. Assuming that $\frac{\partial (c_s h)}{\partial t} = 0$ (an important model assumption that potentially restricts

applicability to hydrograph-scale modeling), the spatial gradient in sediment flux is a balance between sediment entrainment, rock erosion, and sediment deposition:

$$\frac{\partial \left(Q_s/w\right)}{\partial x} = E_s + \left(1 - F_f\right) E_r - D_s. \tag{3}$$

### 4.2   Conservation of sediment and rock on the channel bed

Change of channel bed elevation with time is the sum of changes in rock height and sediment thickness:

$$\frac{\partial \eta}{\partial t} = \frac{\partial R}{\partial t} + \frac{\partial H}{\partial t}. \tag{4}$$

Following Eq. (4) and Davy and Lague (2009), conservation of sediment on the channel bed with sediment thickness $H$ may be written as

$$\frac{\partial H}{\partial t} = \frac{D_s - E_s}{1 - \phi}, \tag{5}$$

where $\phi$ is the porosity of the bed sediment. Because there is no deposition of bedrock, the rate of change of bedrock elevation

$R$ over time is

$$\frac{\partial R}{\partial t} = U - E_r \tag{6}$$

where $U$ is the rock uplift rate relative to baselevel. For simplicity we assume that the porosity of rock is zero, but rock porosity could easily be added to Eq. (6). Steady sediment thickness requires equal erosion and deposition of sediment ($D_s = E_s$) and steady bedrock elevation requires that erosion of rock is balanced by rock uplift relative to baselevel ($U = E_r$). Figure 1 shows

a schematic of a model cell and defines relevant variables.

### 4.3   Entrainment of bed sediment and erosion of bedrock

As discussed by Davy and Lague (2009), any number of accepted expressions for entrainment of bed sediment and erosion of bedrock could be used for $E_s$ and $E_r$, respectively. We consider here the unit stream power formulation in which entrainment

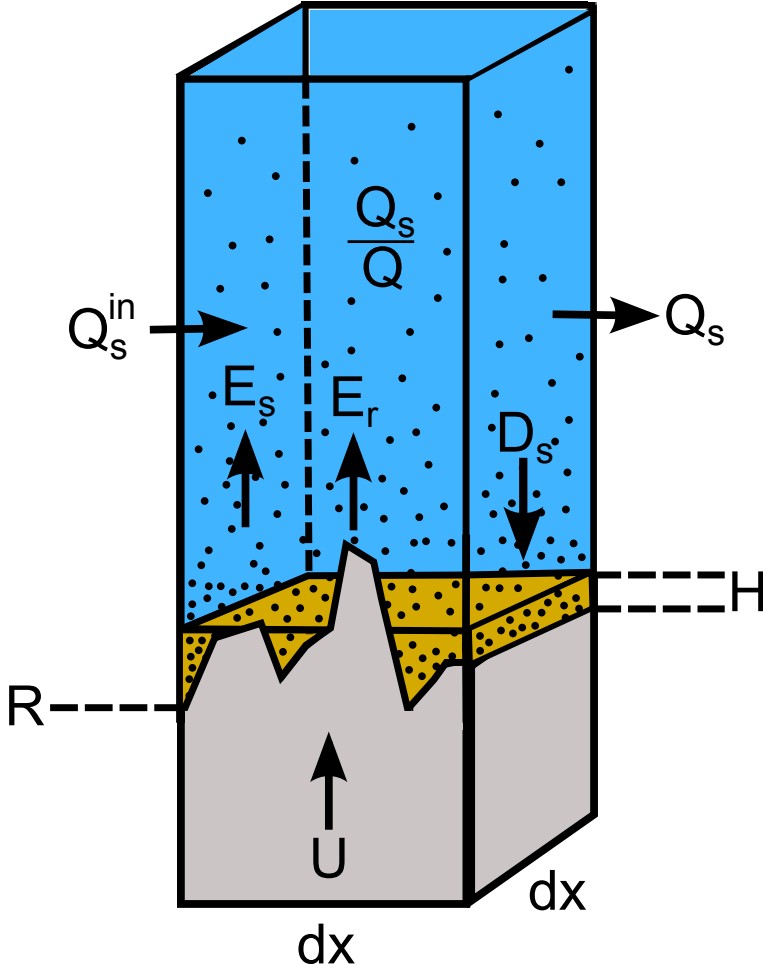

**Figure 1.** Conceptual sketch of a single model cell and definition of variables. Rock surface elevation $R$ is defined as the elevation of the lowest point on the bedrock surface, and is also used as the lower boundary for the thickness $H$ of the alluvial layer.

and erosion are functions of water flux and slope (Howard, 1994; Whipple and Tucker, 1999). We add to the unit stream power model a means by which the flow may simultaneously entrain sediment and erode bedrock, and by which the bed may smoothly transition from fully alluviated to bare bedrock. Consider that the volumetric entrainment rate per unit bed area of bed sediment depends on unit stream power, but is also influenced by sediment thickness $H$ relative to a reach-scale bedrock roughness length scale $H_*$. At low $H/H_*$, the lower points on the bedrock bed are mantled with sediment, but high points on the bedrock surface are still exposed. At high $H/H_*$, all areas of the bed are covered with sediment. Further, $E_s$ must be zero when $H$ is zero and no sediment is available to be entrained, and $E_s$ must reach a maximum (for a given stream power) when there is enough sediment to fully cover the bed (Fig. 2). This approach is conceptually similar to the models of Turowski et al. (2007), Lague (2010) (his exponential model), and Zhang et al. (2015), and similarly eliminates the need to explicitly track

bedrock exposure. Given the conceptual model described above, the entrainment rate of sediment $E_s$ may be written as:

$$E_s = \left(K_s q S^n - \omega_{cs}\right)\left(1 - e^{-H/H_*}\right), \tag{7}$$

where $K_s$ is the sediment erodibility parameter, $S$ is channel-bed slope, and $n$ is a scaling exponent. $q$ is water discharge per unit channel width, and may be calculated by any number of methods. The simplest is $q = k_q A^m$ where $A$ is drainage area, $m$ is a scaling exponent (generally $\approx 0.5$) designed to reflect downstream width changes and discharge–area scaling (Leopold and Maddock, 1953; Snyder et al., 2003b; Wohl and David, 2008), and $k_q$ is an empirical constant, with units dependent on the value of $m$, that is subsumed into $K_s$ and the rock erodibility parameter $K_r$. Using the volumetric water flux per unit channel width $q$ results in $K_s$ and $K_r$ having dimensions of $\left[L^{-1}\right]$. $\omega_{cs}$ is a threshold stream power required for sediment entrainment. In very simple cases $\omega_{cs}$ may be neglected, but the ability to include a threshold term is important in any river evolution model as threshold effects have been shown to significantly alter model outcomes (Snyder et al., 2003a; Tucker, 2004; Lague et al., 2005; DiBiase and Whipple, 2011). While we employ a simple closure for channel width in which width scales as the square root of water discharge (e.g., Leopold and Maddock, 1953; Wohl and David, 2008), it may be desirable for some applications to add dynamic channel width adjustments to the model, as previous work has suggested that width trades off with slope in transient channels (e.g., Finnegan et al., 2005; Turowski et al., 2006; Wobus et al., 2006; Whittaker et al., 2007; Attal et al., 2008; Lague, 2010; Yanites and Tucker, 2010). One option for incorporating dynamic width is to calculate or approximate shear-stress distributions across channel cross-sections (e.g., Kean and Smith, 2004; Wobus et al., 2006, 2008; Turowski et al, 2009). A simpler dynamic width rule can be obtained by partitioning erosive power between the bed and banks under a trapezoidal channel assumption (Flintham and Carling, 1988) as detailed by Lague (2010). Different approaches have different numbers of parameters and computational costs, and further work will be necessary to elucidate which advances beyond the standard empirical width closure are tractable within the SPACE landscape evolution model framework.

If sediment entrainment declines with decreasing sediment thickness as a result of increased bedrock exposure, bedrock erosion should follow an inverse but conceptually similar pattern (Fig. 2). Assuming that increasing mean sediment thickness leads to higher proportions of bedrock covered by sediment and that sediment cover inhibits erosion (e.g., Beaumont et al., 1992; Whipple and Tucker, 2002; Sklar and Dietrich, 2004; Gasparini et al., 2006, 2007; Turowski et al., 2007; Hobley et al., 2011), the volumetric erosion rate of bedrock per unit bed area may be written as

$$E_r = \left(K_r q S^n - \omega_{cr}\right) e^{-H/H_*}. \tag{8}$$

Here, $K_r$ is the bedrock erodibility parameter, which is generally expected to be substantially lower than $K_s$. $\omega_{cr}$ is the threshold stream power for detachment of bedrock, which may vary significantly depending on the relative dominance of plucking or abrasion (e.g., Hancock et al., 1998; Whipple et al., 2000) as well as the weathered state of the bedrock (Hancock et al., 2011; Johnson and Finnegan, 2015; Small et al., 2015; Murphy et al., 2016; Shobe et al., 2017). Howard (1998), Hancock and Anderson (2002), Turowski et al. (2007), and Lague (2010) employed a similar exponential decline in rock erosion rate with increasing sediment thickness. Eq. (8) falls into the category of "cover" models, which treat erosion reduction by sediment shielding the bed without incorporating the potential erosive effects of mobile sediment (e.g., Beaumont et al., 1992; Lague,

2010; Shobe et al., 2016). The smooth transitions between bare-bedrock, bedrock-alluvial, and fully alluviated channels given by Eq. (7 and 8) are both more stable and more realistic than models in which the existence of any alluvium fully covers the bedrock (implying a perfectly smooth, planar bedrock surface). Figure 2 shows the pattern of sediment entrainment and bedrock erosion over different values of $H/H_*$. $f(H/H_*)$ in Fig. 2 is the dimensionless exposure term that modifies stream power entrainment and erosion in Eq. (7 and 8) (i.e., $e^{-H/H_*}$ or $\left(1 - e^{-H/H_*}\right)$). Perhaps the most significant simplification in our model is that we do not include explicit treatment of the erosive effects of grains in transport (e.g., Sklar and Dietrich, 1998, 2001, 2004; Gasparini et al., 2006; Turowski et al., 2007; Lamb et al., 2008; Cook et al., 2013). Such an effect could enhance sediment entrainment if grains in saltation hit resting grains and enabled their entrainment, and could enhance bedrock erosion if sediment-rich water were flowing over well-exposed bedrock. The "tools effect" on bedrock erosion could be incorporated into the model by assuming that $E_r$ at a given $H/H_*$ increases with $Q_s$ until $Q_s$ reaches transport capacity (which is $Q_s$ when $E_s + E_r = D_s$). Because increases in $H/H_*$ already account for the "cover effect," the $E_r$ dependence on $Q_s$ need only be positive and not decline with increasing $Q_s$ (e.g., Sklar and Dietrich, 2001; Gasparini et al., 2006; Turowski et al., 2007). We assume for the purposes of model validation against analytical solutions that these effects are negligible relative to changes in unit stream power and bed cover.

The major advantages of the exponential entrainment and erosion approach outlined here are that 1) sediment and bedrock may be simultaneously entrained/eroded into the water column, 2) the presence of sediment does not completely inhibit bedrock erosion at low values of $H/H_*$, which is supported by modeling and observations of mixed bedrock-alluvial channels (Johnson et al., 2009; Johnson, 2014; Ferguson et al., 2017; Hodge, 2017) and 3) model stability is improved because sediment thickness gradually approaches zero, preventing a sudden transition from sediment entrainment to bedrock erosion. Many different rules for sediment entrainment and bedrock erosion could be used in this model framework in place of stream-power type equations. No matter how erosive power is calculated, the SPACE approach allows both erodibility and entrainment/detachment thresholds to be chosen independently for sediment and bedrock, unlike in strict detachment/transport limited models or the basic form of erosion-deposition models. This enables the treatment of systems with multiple erosion thresholds, such as a river for which bedrock erosion requires both mobilization of an alluvial cover (described by $\omega_{cs}$) and the plucking of bedrock blocks (described by $\omega_{cr}$).

### 4.3.1 Optional smoothing of entrainment/erosion thresholds

The approach outlined above allows for the incorporation of an entrainment threshold for sediment and an erosion threshold for bedrock such that entrainment/erosion is zero when stream power is below the chosen threshold(s). Erosion thresholds representing a sharp transition between no erosion (below threshold) and erosion (above threshold) have a long history in fluvial erosion modeling (e.g., Snyder et al., 2003a; Tucker, 2004; Lague et al., 2005; DiBiase and Whipple, 2011). With a sharp or discontinuous erosion threshold, the transition from no erosion to erosion is abrupt. However, the sediment transport literature suggests that sediment entrainment is often better-represented by a distribution of entrainment thresholds than a single threshold value (Kirchner et al., 1990; Wilcock and McArdell, 1997; McEwan and Heald, 2001). Using a distribution of entrainment thresholds helps to account for the observation that incipient sediment motion does not begin at the same

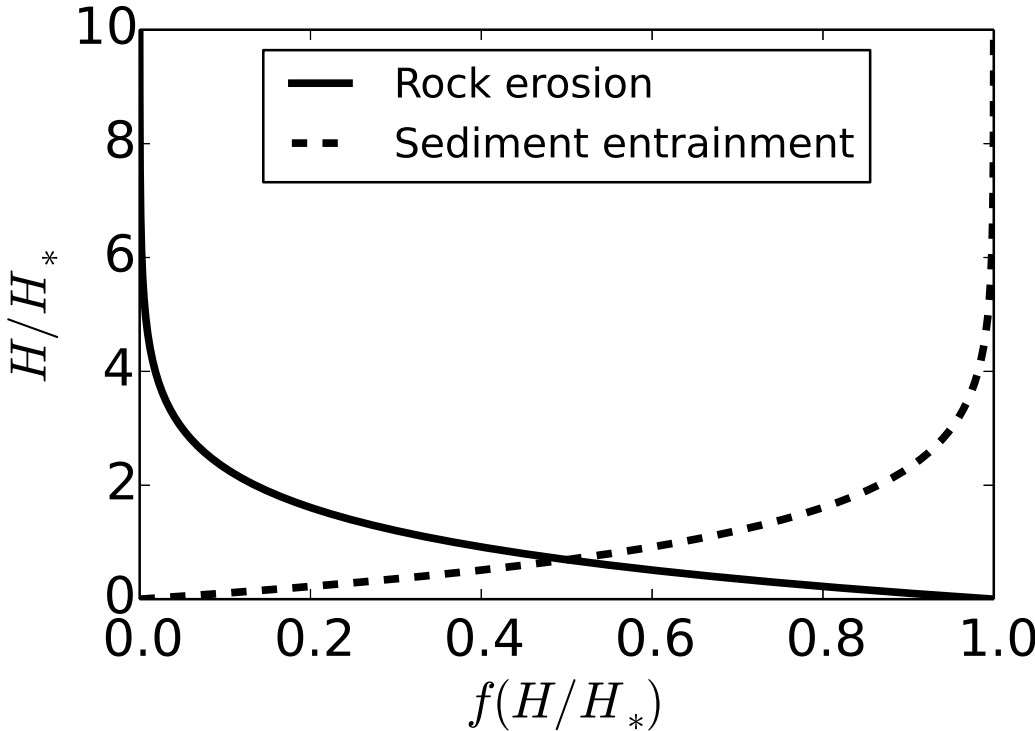

**Figure 2.** Dimensionless efficiency of erosion and deposition ($f(H/H_*)$) for different values of $H/H_*$. Erosive power is multiplied by $f(H/H_*)$ in the model to account for the relative exposure of sediment and bedrock. Such a formulation accounts for the fact that bedrock beds are rough, and low points may become sediment-mantled while high points remain exposed (e.g., Johnson, 2014; Zhang et al., 2015). $H_*$ is therefore a length scale representing reach-scale bedrock roughness. Sediment entrainment for a given stream power increases with increasing $H/H_*$, while bedrock erosion declines as a response to sediment mantling of the bed. At values of $H/H_* \approx 6$, all bed lowering is driven by sediment entrainment and bedrock erosion is negligible. This heuristic representation of mixed alluvial-bedrock channel dynamics is conceptually similar to the approaches taken by Lague (2010) and Zhang et al. (2015).

threshold value for all particles (e.g., Buffington and Montgomery, 1997). Variability in the threshold for motion is thought to be a function of grain size and shape variations (Kirchner et al., 1990; Wilcock and McArdell, 1997; Prancevic and Lamb, 2015), grain hiding and protrusion effects (Kirchner et al., 1990; Parker, 1990; Wilcock and McArdell, 1997; McEwan and Heald, 2001), bed sorting (Nelson et al., 2009), sediment flux (Johnson, 2016), and flow history (Masteller and Finnegan, 2017). Similarly, the threshold for bedrock erosion can depend on sub-reach scale mineralogy, joint spacing and orientation, and the weathered state of the bedrock. Given that the SPACE model operates on too large a scale to treat such processes explicitly, we develop a formulation for entrainment and erosion thresholds that reflects the variability in the threshold. If a distribution of thresholds exists, erosion should decline to zero not exactly when available stream power drops below the user-defined threshold as would be the case for a standard threshold model, but when available stream power is significantly

below the defined threshold. As available stream power becomes larger than the defined threshold, entrainment and erosion should increase smoothly as a greater portion of the distribution of thresholds is exceeded. In the limit where available stream power is many times greater than the user-defined threshold, available stream power should simply be reduced by the user-defined threshold. An exponential function describing the increase in entrainment/erosion as available stream power increases

relative to threshold stream power satisfies these requirements without adding any model parameters. We include an optional exponential expression for threshold stream power such that entrainment/erosion does not go immediately to zero when stream power $\omega$ equals the threshold value $\omega_c$, but declines exponentially as $\omega/\omega_c$ declines. The threshold stream power is expected to be different for rock than for sediment, with $\omega_{cr}$ likely being larger in most cases than $\omega_{cs}$. In this formulation, the expressions for sediment entrainment and bedrock erosion (Eq. (7 and 8)) become:

$$E_s = \left( K_s q S^n - \omega_{cs} \left( 1 - e^{-\omega/\omega_{cs}} \right) \right) \left( 1 - e^{-H/H_*} \right) \tag{9}$$

and

$$E_r = \left( K_r q S^n - \omega_{cr} \left( 1 - e^{-\omega/\omega_{cr}} \right) \right) e^{-H/H_*}. \tag{10}$$

Inspection of Eq. (9 and 10) reveals that when $\omega \gg \omega_c$, the threshold term approaches $\omega_c$, yielding behavior identical to single-value threshold models. When $\omega = \omega_c$, the threshold represents approximately 63% $(1 - e^{-1})$ of available stream power, rather

than 100% as in the basic threshold approach. When $\omega \ll \omega_c$, the threshold term approaches $\omega$ and the entrainment or erosion rate approaches zero. We chose an exponential function because it allows for smoothing of entrainment and erosion thresholds, and therefore honors the reality that such thresholds tend to be distributions of values rather than a single value, without adding any model parameters. Evaluation of the full behavior of models using an exponentially declining threshold is beyond the scope of this paper, and the use of Eq. (9 and 10) is optional in the SPACE model.

**4.4  Deposition of sediment**

The flux of sediment from the water column onto the bed is the product of sediment concentration averaged over the depth of the water column and effective sediment settling velocity $V$ (Davy and Lague, 2009):

$$D_s = c_s V = \frac{Q_s}{Q} V. \tag{11}$$

$V$ is not the still-water particle settling velocity, but is the net effective settling velocity after accounting for the upward effects

of turbulence. $V$ also incorporates the vertical gradient in sediment concentration through the water column ($d^*$ in Davy and Lague (2009)). In an equivalent formulation, Davy and Lague (2009) treated the sediment deposition rate as $D_s = d^* c_s V$, where $d^*$ is a dimensionless number that relates sediment concentration near the bed to mean sediment concentration in the water column. Eq. (11) assumes that sediment and water move at the same speed such that all changes in $\frac{Q_s}{Q}$ are driven by erosion and deposition.

## 4.5 Steady-state analytical solutions

We develop steady-state analytical solutions for sediment flux $Q_s$, channel slope $S$, and bed sediment thickness $H$, all of which are steady when $\frac{\partial(c_s h)}{\partial t} = 0$. We assume for the purposes of this derivation that there are no entrainment or erosion thresholds, and that $F_f$ and $\phi$ are both negligible, assumptions that are easily relaxed. We define steady state in this system as a state of
time-invariant bedrock elevation and topographic elevation (which also implies time-invariant sediment thickness). This occurs when two conditions are satisfied. First, rock uplift must be balanced by bedrock erosion such that

$$\frac{\partial R}{\partial t} = 0 = U - K_r q S^n e^{-H/H_*}. \tag{12}$$

Second, sediment entrainment and deposition must balance each other such that sediment thickness $H$ is unchanging in time:

$$\frac{\partial H}{\partial t} = 0 = V\frac{Q_s}{Q} - K_s q S^n \left(1 - e^{-H/H_*}\right). \tag{13}$$

At steady state, the volumetric sediment flux $Q_s$ at any point along the channel must balance the volume of newly uplifted rock in the area draining to that point:

$$Q_s = UA. \tag{14}$$

To find steady-state channel slope, we begin by rearranging Eq. (13) and combining with Eq. (14):

$$K_s q S^n \left(1 - e^{-H/H_*}\right) = V\frac{UA}{Q}. \tag{15}$$

Recognizing that $Q = Ar$ where $r$ is a runoff rate per unit area:

$$K_s q S^n \left(1 - e^{-H/H_*}\right) = \frac{VU}{r}. \tag{16}$$

We rearrange Eq. (16) to isolate $e^{-H/H_*}$, substitute into Eq. (12) and solve for $S$ to yield:

$$S = \left[\frac{UV}{K_s q r} + \frac{U}{K_r q}\right]^{1/n}, \tag{17}$$

or if $q = k_q A^m$ as in the simple stream power formulation,

$$S = \left[\frac{UV}{K_s A^m r} + \frac{U}{K_r A^m}\right]^{1/n}, \tag{18}$$

where $k_q$ is subsumed into $K_s$ and $K_r$. When $n = 1$, the second term on the right-hand side is the slope predicted by a detachment-limited incision model in which slope increases with faster rock uplift, lower rock erodibility, or less water discharge (or drainage area). The first term on the right-hand side describes the additional component of slope required to transport sediment. That component of slope must increase with increasing settling velocity, lower sediment erodibility, and lower water
discharge. Davy and Lague (2009) derived a similar expression for slope-discharge scaling for their erosion-deposition model. The major difference between our result and theirs is that their expression is for slope of a single bed material with a single

bed erodibility when erosion balances rock uplift, whereas Eq. (18) incorporates equilibrium in both sediment thickness and bedrock height. Eq. (18) may be rearranged to show that SPACE predicts a standard stream power slope-area relationship modulated by $\frac{V}{r}$ as well as sediment and bedrock erodibility:

$$S = \left[ \frac{V}{K_s r} + \frac{1}{K_r} \right]^{1/n} U^{1/n} A^{-m/n}. \tag{19}$$

The ratio between the effective settling velocity $V$ and the runoff rate $r$ controls the relative importance of the bedrock and alluvial components of the steady-state channel slope. In the simplified case of $K_s = K_r$, a ratio of $\frac{V}{r} = 1$ would indicate equal contributions from the two regimes. Quantifying $\frac{V}{r}$ for natural systems could therefore give a valuable indication of process dynamics in natural channels.

Solving Eq. (16) for $H$ gives steady-state bed sediment thickness as a function of channel slope:

$$H = -H_* \ln \left[ 1 - \frac{VU}{rK_s qS^n} \right]. \tag{20}$$

To obtain a slope-independent solution for $H$, we can combine Eq. (18 and 20) and simplify:

$$H = -H_* \ln \left[ 1 - \frac{V}{\frac{K_s r}{K_r} + V} \right]. \tag{21}$$

The SPACE model therefore predicts constant sediment thickness along the channel at steady state as long as all parameters in Eq. (21) are constant in space. As settling velocity becomes larger, $\frac{V}{\frac{K_s r}{K_r} + V}$ approaches one and $H$ becomes large. As $K_s$

increases and sediment is more easily entrained from the bed, $\frac{V}{\frac{K_s r}{K_r} + V}$ and therefore $H$ both approach zero. Increasing bedrock erodibility $K_r$ causes an increase in steady state $H$ as more sediment is created from detached bedrock.

## 4.6    Dimensional analysis

We present a nondimensionalization of the model described above. For simplicity, we assume that sediment entrainment and bedrock erosion thresholds are negligible, though the model allows independent entrainment and erosion thresholds as shown

above. The model contains three independent variables, $Q_s$, $H$, and $R$, the latter two of which are summed to give land surface elevation. Each of these variables requires a scale for nondimensionalization. We begin by nondimensionalizing sediment flux:

$$Q_s' = \frac{Q_s V}{K_s q^2 S w}. \tag{22}$$

Sediment thickness $H$ and bedrock elevation $R$ may both be scaled by the sediment layer length-scale $H_*$:

$$H' = H/H_* \tag{23}$$

and

$$R' = R/H_*, \tag{24}$$

and downstream distance $x$ by the length scale $q/V$ (noted by Davy and Lague (2009) and found to govern the transition between detachment-limited and transport-limited behavior):

$$x' = xV/q. \tag{25}$$

Finally, time $t$ is nondimensionalized by:

5    $$t' = tV/H_*. \tag{26}$$

Replacing the dimensionless variables into the governing equations yields the following equations for dimensionless sediment flux, sediment thickness, and rock elevation, which are applicable for negligible erosion thresholds:

$$\frac{\partial Q_s'}{\partial x'} = S\left(1 - e^{-H'}\right) + (1 - F_f)\left[\frac{K_r}{K_s}\right] S e^{-H'} - Q_s' \tag{27}$$

10    $$\frac{\partial H'}{\partial t'} = \left[\frac{K_s q}{V}\right]\left(Q_s' - S\left(1 - e^{-H'}\right)\right) \tag{28}$$

$$\frac{\partial R'}{\partial t'} = \left[\frac{U}{V}\right] - \left[\frac{K_r}{K_s}\right]\left[\frac{K_s q}{V}\right] S e^{-H'}. \tag{29}$$

Three dimensionless parameters appear in Eq. (27–29): a normalized rock uplift rate $\left[\frac{U}{V}\right]$, a ratio of erodibilities $\left[\frac{K_r}{K_s}\right]$, and a sediment entrainment ratio $\left[\frac{K_s q}{V}\right]$. The normalized rock uplift rate shows the relative importance of rock uplift and grain 15   settling velocity, with increases in both driving increased channel slope. The erodibility ratio reflects the relative ease of eroding bedrock and entraining sediment, and is influenced in natural systems by bedrock and sediment lithology, grain size, and grain sorting on the channel bed. The sediment entrainment ratio encompasses competition between sediment entrainment, which is driven by high water discharge and high sediment erodibility, and sediment deposition, driven by high grain settling velocity. Notably, the sediment entrainment ratio contains the $q/V$ length scale described by Davy and Lague (2009) as the average 20   travel distance of sediment grains from entrainment to re-deposition. The model predicts detachment-limited behavior when $q/V$, and therefore our entrainment ratio, are large, and transport-limited behavior when they are small. Specifically, following Davy and Lague (2009), we can define a dimensionless number $\frac{V}{r}$ that governs the transition between detachment-limited and transport-limited dynamics. In the sediment-only case (when $H \gg H_*$), or in the bedrock-only case ($H = 0$), $\frac{V}{r} > 1$ gives transport-limited behavior and $\frac{V}{r} < 1$ results in detachment-limited behavior (Davy and Lague, 2009). In cases where sediment 25   and bedrock are eroded simultaneously, especially if there is a significant erodibility contrast between the two, the behavior is not so easily predicted and will generally contain contributions from both detachment and transport limitations.

## 5   Numerical implementation and local analytical solutions

In this section we describe the forward-time numerical solution of the SPACE model in two dimensions. The solution to the model equations in each timestep consists of three conceptual steps. First, sediment flux is calculated with a local analytical

solution, described below, at every node working in order from upstream to downstream. Second, sediment thickness is calculated at every node using a local analytical solution for $H(t)$, which we develop below. Third, bedrock erosion is calculated for each node.

## 5.1 Calculation of sediment flux

As shown in Eq. (3), the $x$-directed rate of change in sediment flux depends on sediment entrainment, bedrock erosion, and sediment deposition. The dependence of deposition rate on $\frac{Q_s}{Q}$ means that the deposition flux, and therefore the change in sediment flux, at a given node depends on the sediment flux entering that node from upstream (Eq. 11). It is therefore critical to order nodes in upstream to downstream order and calculate sediment flux iteratively from upstream to downstream. This approach unfortunately precludes simultaneous calculation of sediment flux at all nodes. Landlab's flow routing capabilities order all nodes into a "stack" following the methodology of Braun and Willett (2013). Because our sediment flux calculations must progress from upstream to downstream, we use their "inverted stack order" in which nodes are ordered from upstream to downstream, allowing the SPACE algorithm to efficiently sum water and sediment fluxes at tributary junctions. In addition, the downstream sediment flux calculation is written and compiled using the Cython library, giving it significant performance improvements over the same loop in pure Python.

Numerical integrations of sediment entrainment and deposition are often significant sources of model inaccuracy and instability due to the spatial extrapolation of linear entrainment and deposition equations. Consider a river reach of length $\mathrm{d}x$ with clear water entering the reach at the upstream end. The initial sediment entrainment rate is $E_s = K_s q S^n - \omega_{cs}$ and the initial deposition rate is zero because $Q_s = 0$. However, two natural processes make the linear extrapolation of these initial entrainment and deposition rates over cell length $\mathrm{d}x$ inappropriate. First, sediment entrainment rate may decline over $x$ as sediment thickness $H$ declines if $H$ is not much greater than $H_*$. Second, as sediment is entrained over distance $\mathrm{d}x$, $Q_s$ increases, which drives a progressive increase in deposition rate. Simply numerically integrating Eq. (3) to calculate sediment flux does not account for either the progressive decline in available sediment or the progressive saturation of the water column and increase in deposition flux. We have therefore developed a local analytical solution to account for such effects. This approach prevents severe overestimation of sediment entrainment into the water column, making SPACE more stable than models that do not account for within-cell changes in $Q_s$. Our local analytical solution for sediment flux accounts for the fact that sediment flux from upstream $Q_s^{\mathrm{in}}$ and any net erosion (or deposition) in a model cell of area $\mathrm{d}x^2$ contribute to $Q_s$, which drives deposition. Let $Q_s^{\mathrm{out}}$ represent the sum of sediment influx, erosion, and deposition such that $Q_s^{\mathrm{out}}$ is the net erosion rate in the cell multiplied by the cell area:

$$Q_s^{\mathrm{out}} = Q_s^{\mathrm{in}} + (1 - \phi) E_s \mathrm{d}x^2 + (1 - F_f) E_r \mathrm{d}x^2 - V \frac{Q_s}{rA}. \tag{30}$$

Because sediment deposition in a cell depends on both $Q_s^{\mathrm{in}}$ from upstream and sediment entrained from the cell itself, we can substitute $Q_s^{\mathrm{out}}$ for $Q_s$ in the deposition term. Eq. (30) may then be solved to yield the local analytical solution for $Q_s$ within a model cell:

$$Q_s^{\mathrm{out}} = \frac{Q_s^{\mathrm{in}} + (1 - \phi) E_s \mathrm{d}x^2 + (1 - F_f) E_r \mathrm{d}x^2}{1 + V \mathrm{d}x^2 / (rA)}. \tag{31}$$

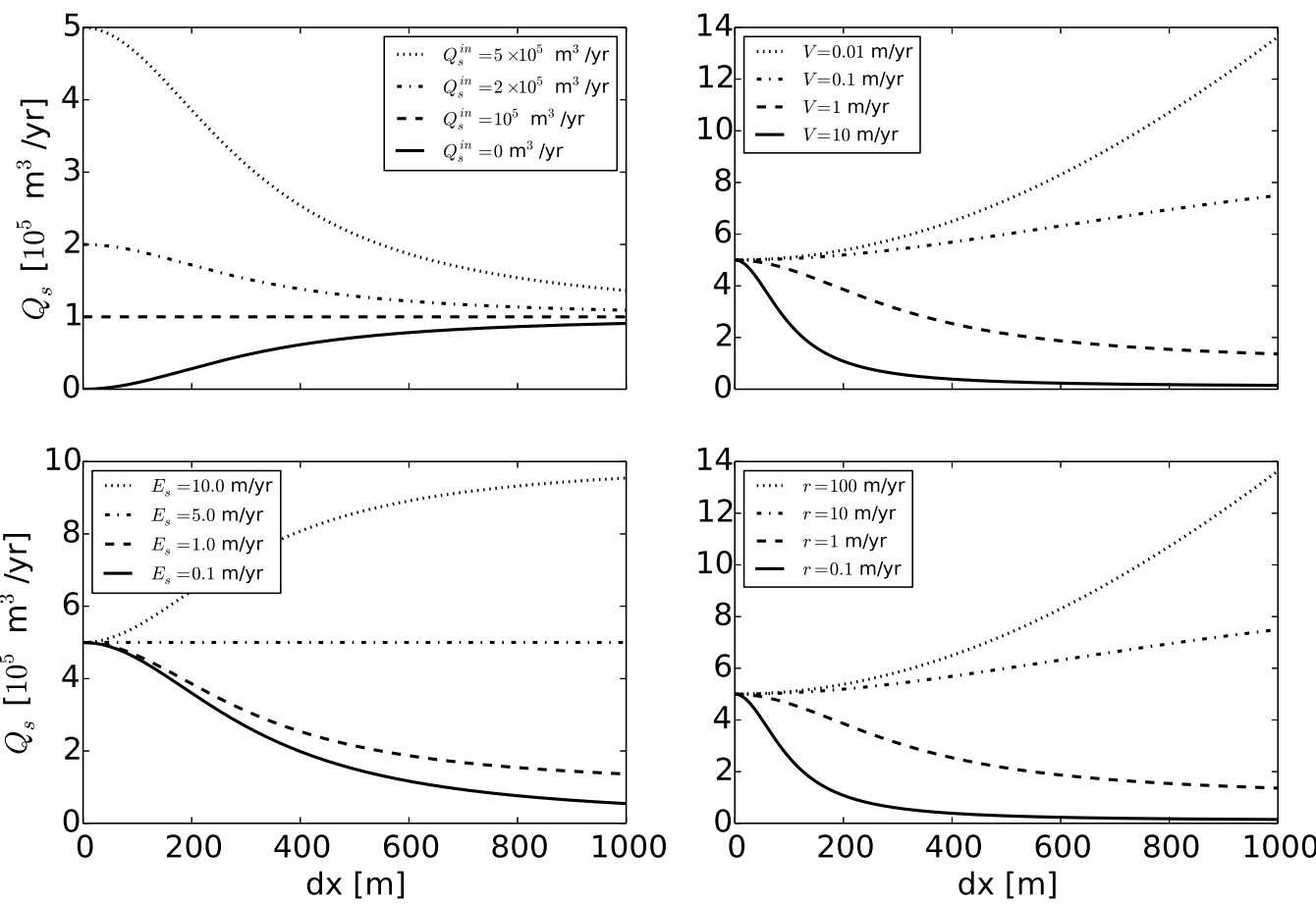

**Figure 3.** Sediment flux $Q_s$ as a function of distance $\mathrm{d}x$ as calculated by the local analytical solution (Eq. (31)) at a given drainage area ($A = 10^5 \ \mathrm{m}^2$). In each panel, one parameter is varied while all others are held constant. The sediment flux coming in from upstream is an important control on $Q_s$ at short length scales, but declines in importance as $\mathrm{d}x$ approaches 1000 m. High values of settling velocity cause low $Q_s$ and vice versa. High sediment entrainment rates lead to high $Q_s$ as do high runoff rates. Parameter values (except where changed in the four panels) are: $Q_s^{in} = 50,000 \ \mathrm{m}^3/\mathrm{yr}$, $V = 1.0 \ \mathrm{m/yr}$, $E_s = 1.0 \ \mathrm{m/yr}$, and $r = 1.0 \ \mathrm{m/yr}$. $E_r = 0$ and $\phi = 0$ for simplicity in this case.

Eq. (31) breaks down where $rA = 0$, which is acceptable because $Q_s$ will always be zero where $rA = 0$. Figure 3 shows $Q_s$ as a function of some of the relevant variables in Eq. (31).

## 5.2 Calculation of sediment thickness

After calculation of $Q_s$ at every node, sediment thickness $H(t)$ is calculated according to Eq. (5). Similar to the solution of the sediment flux equation, solution of $H(t)$ is subject to numerical inaccuracies and instabilities driven by the dependence of $E_s$ on $H$. Extrapolating $\frac{\partial H}{\partial t}$ over a full timestep using $H_0$ ($H$ at the beginning of the timestep) causes overestimation of sediment

entrainment, especially at larger timesteps. We therefore develop a local analytical solution for $H(t)$ for a small time interval over which variations in $H$ are important but $D_s$, $K_s$ $q$, and $S$ (as well as any entrainment threshold) may be considered steady. We find the analytical solution for $H(t)$ by integrating Eq. (5) with respect to time with the knowledge that $H$ has some initial value $H_0$ at the beginning of a timestep ($t = 0$):

$$H(t) = H_* \ln \left[ \frac{1}{(D_s/(1-\phi))/\widehat{E}_s - 1} \left( e^{\left( D_s/(1-\phi) - \widehat{E}_s \right) t/H_*} \left( \left( \frac{(D_s/(1-\phi))}{\widehat{E}_s} - 1 \right) e^{H_0/H_*} + 1 \right) - 1 \right) \right], \tag{32}$$

where

$$\widehat{E}_s = K_s q S^n - \omega_{cs}. \tag{33}$$

Inspection of Eq. (32) reveals that $H(t)$ may become undefined in two physically realistic situations. The first is where $(D_s/(1-\phi))/\widehat{E}_s = 1$ and the second is where $\widehat{E}_s = 0$. Eq. (32) is therefore only applied at nodes with $D_s/(1-\phi) \neq \widehat{E}_s$ and $\widehat{E}_s > 0$. When $D_s/(1-\phi) = \widehat{E}_s$, the change in alluvium thickness with time becomes $\frac{\partial H}{\partial t} = e^{-H/H_*}$. Integrating with respect to time and applying $H = H_0$ at $t = 0$ gives the solution for $H(t)$ when $D_s/(1-\phi) = \widehat{E}_s$:

$$H(t) = H_* \ln \left[ \frac{K_s q S^n - \omega_{cs}}{H_*} t + e^{H_0/H_*} \right]. \tag{34}$$

When $\widehat{E}_s \leq 0$, no entrainment of sediment occurs and any changes in $H(t)$ are driven by deposition. In this case, changes in $H$ are computed with a simple forward numerical solution:

$$H(t) = H_0 + \frac{D_s}{1-\phi} \mathrm{d}t. \tag{35}$$

In all cases, the relevant equation for $H(t)$ is solved each timestep using $t = \mathrm{d}t$ where $\mathrm{d}t$ is the model timestep length. Unlike for $Q_s$, $H$ may be simultaneously calculated at every node, allowing efficient solution of Eq. (32–35) over the entire model domain. Figure 4 shows $H(t)$ as a function of some of the relevant variables in Eq. (32).

### 5.3 Calculation of change in bedrock height

Bedrock erosion is calculated by combining Eq. (6 and 8) and solving forward in time from a previous rock elevation $R_0$:

$$R = R_0 + \left( U - \left( K_r q S^n - \omega_{cr} \right) e^{-H/H_*} \right) \mathrm{d}t. \tag{36}$$

The simple forward numerical solution employed in Eq. (36) becomes inappropriate at very large timesteps, as $H$ may change significantly, influencing channel slopes and therefore bedrock erosion. However, because bedrock erosion is generally a much slower process than sediment entrainment in most cases, Eq. (36) is unlikely to introduce substantial instability.

## 6 Implementing SPACE in Landlab

### 6.1 Landlab modeling toolkit

Landlab is a flexible, open-source modeling framework written in Python that allows efficient model building and hypothesis testing across many subdisciplines in Earth system science (Hobley et al., 2017). Landlab is a plug-and-play environment in

which users can easily build two-dimensional numerical models consisting of any number of well-vetted components (e.g., Tucker et al., 2016; Adams et al., 2017) along with user-specific equations and functionality. The greatest advantages of using Landlab are 1) its built-in gridding engine, which creates model grids, efficiently stores spatially distributed variables, and handles boundary conditions, and 2) the ability to easily couple different components into a single model sharing a single grid.
Landlab allows efficient coupling of components representing fluvial erosion, hillslope processes, basin hydrology (Adams et al., 2017), geodynamics, vegetation, and many other processes into novel surface dynamics models.

   Landlab's gridding engine supports grids consisting of square and rectangular grids ("raster grids"), hexagonal grids, and Voronoi–Delaunay interlocked meshes (Hobley et al., 2017). Every Landlab grid is made up of nodes, cells, and links. Nodes are points in $(x, y)$ space. Cells are polygons surrounding all non-boundary (interior) nodes that may be rectangular, hexagonal,
or defined by Voronoi polygons depending on the chosen grid type. Links connect adjacent pairs of nodes and are directional. Rectangular grids have four links per node, hexagonal grids have six, and Voronoi grids have a number of links per node equivalent to the number of faces on each Voronoi polygon. Links have default directionality, but this directionality does not determine the directions of fluxes in Landlab models, which are set by gradients along links. In this paper, we focus for simplicity on a square ($\Delta x = \Delta y$) raster grid, which is currently the only grid type supported by the SPACE model. A
diagram of a generic raster grid is shown in Fig. 5. Nodes, cells, and links may all store model data in the form of NumPy arrays associated with one of the three grid elements. Each data field is defined by a keyword in a dictionary data structure attached to a certain grid element. The SPACE model, for example, tracks sediment depth at all grid nodes, a field that may be accessed by any component (this field is called "soil depth" in Landlab to keep terminology standard between hillslope and fluvial components). An array of sediment depths at all grid nodes could be found by typing: *grid.at_node["soil__depth"]*. The
treatment of boundary conditions in Landlab grids is described thoroughly by Hobley et al. (2017) and Adams et al. (2017). In short, nodes may be set as "boundary" nodes and then defined as open, fixed-gradient, or closed boundaries. Non-boundary nodes are set as "core" nodes.

# 7  Verification and evaluation: Comparison to analytical solutions for detachment-limited, transport-limited, and mixed cases

As discussed above, the SPACE model equations are capable of replicating both detachment-limited (when there is no sediment and $F_f = 1$) and transport-limited (when $H \gg H_*$) model behavior. In this section we compare the behavior of the SPACE 1.0 Landlab component (the numerical implementation of the SPACE model equations presented above) to steady-state analytical solutions for standard detachment-limited and transport-limited models to assess whether our numerical implementation of the SPACE algorithm can replicate these two end-member cases. In addition, we test the performance of the SPACE component
against steady-state analytical solutions for a mixed case where both bedrock erosion and sediment transport influence channel evolution. For the three test cases we use a simple 20 x 20 node square raster grid with $\mathrm{d}x = 100$ m, for a 2 km x 2 km model domain. The initial topography of the domain is a plane tilted to the lower-left (southwest) corner with random microscale roughness to force flow convergence. The lower-left corner is the only open boundary, and is therefore the basin outlet in all

cases. Such a setup results in a model domain that drains to the single open boundary node, allowing predictable drainage
network development. The random seed is held constant so that all runs start from the same initial topography. For simplicity
in these test cases, there are no other surface process models (e.g., hillslope models) coupled to the SPACE component. While
the SPACE 1.0 component is stable at 10-year timesteps under most conditions, we use a timestep of 1 year here to maximize

numerical accuracy for comparison with analytical solutions. We run the model for 100,000 years for the detachment and
transport limited comparisons, and 200,000 years for the mixed bedrock-alluvial comparison (see table 1). We define steady
state as having been achieved when every interior (non-boundary) node is lowering at the same rate as the baselevel node to
within $10^{-6}$ m/yr precision, but allow the model to run for the full imposed run time even after steady state has been achieved.

## 7.1   Detachment-limited comparison

With no sediment ($H = 0$ and $c_s = 0$) and $F_f = 1$ (all bedrock eroded becomes wash load and is not included in model
calculations), all changes in bed elevation are driven by changes in bedrock elevation:

$$\frac{\partial \eta}{\partial t} = \frac{\partial R}{\partial t} \tag{37}$$

where

$$\frac{\partial R}{\partial t} = U - K_r q S^n - \omega_{cr}. \tag{38}$$

When $\omega_{cr} = 0$, Eq. (38) is the simple stream power model (Whipple and Tucker, 1999). At topographic steady state when
$\frac{\partial R}{\partial t} = 0$ and $U = K_r q S^n$, the slope at every point in the channel is

$$S = \left(\frac{U}{K_r q}\right)^{1/n}. \tag{39}$$

Because we use a simple stream power formulation where $q = k_q A^m$ for our test case, the slope-discharge relationship may be
re-written to yield a slope-area relationship:

$$S = \left(\frac{U}{K_r A^m}\right)^{1/n}, \tag{40}$$

where $k_q$ is subsumed into $K_r$. We test whether the SPACE component can replicate steady-state detachment-limited behavior
by comparing slope-area relationships predicted by Eq. (40) with those calculated by the SPACE component. See table 1 for
the parameter values used.

Figure 6 shows the results after the test model domain has achieved topographic steady state. The top panel of Fig. 6 shows

the longitudinal profile of the longest drainage path in the model domain. As predicted by the theory described above, slope
and drainage area trade off such that the outcome is a concave-up longitudinal profile with constant concavity. The lower
panel of Fig. 6 compares the slope-area relationship predicted by Eq. (40) (gray dashed line) to the slope-area relationship
in the steady-state model landscape (black dots). All core nodes from the model domain are shown, and every node obeys
the predicted detachment-limited slope-area scaling. The slope of the slope-area power-law scaling relationship (Fig. 6) is the

channel concavity, thus confirming that the channel concavity observed in the longitudinal profile is constant, and the SPACE
component agrees with theoretical predictions for detachment-limited rivers at steady state.

**Table 1.** Parameter values for SPACE model test cases.

| | Detachment Limited | Transport Limited | Mixed | Coupled |
|---|---|---|---|---|
| Number of rows (-) | 20 | 20 | 20 | 50 |
| Number of columns (-) | 20 | 20 | 20 | 50 |
| Node spacing (m) | 100 | 100 | 100 | 100 |
| Timestep (yr) | 1 | 1 | 1 | 1 |
| Run time (kyr) | 100 | 100 | 200 | 300 |
| Initial $H$ (m) | 0 | 100 | 0 | 0 |
| $U$ (m/yr) | 0.0001 | 0.0001 | 0.0001 | see text |
| $K_r$ (m$^{-1}$) | 0.001 | 0.0001 | 0.005 | 0.0001 |
| $K_s$ (m$^{-1}$) | 0.01 | 0.01 | 0.01 | 0.0005 |
| $m$ (-) | 0.5 | 0.5 | 0.5 | 0.5 |
| $n$ (-) | 1.0 | 1.0 | 1.0 | 1.0 |
| $\omega_{cr}$ (m/yr) | 0 | * | 0 | 0 |
| $\omega_{cs}$ (m/yr) | 0 | 0 | 0 | 0 |
| $H_*$ (m) | 1.0 | 1.0 | 1.0 | 1.0 |
| $\phi$ (-) | 0 | 0 | 0 | 0 |
| $F_f$ (-) | 1 | 0 | 0 | 0 |
| $V$ (m/yr) | 1.0 | 5.0 | 5.0 | 2.0 |

Not all parameters will influence the model outcome in all cases. For example, the value of $V$ is irrelevant for the detachment-limited case when all eroded bedrock passes out of the model domain as permanently suspended fine sediment ($F_f = 1$).

## 7.2 Transport-limited comparison

When sediment thickness $H$ is large relative to $H_*$, changes in bed elevation are driven entirely by changes in sediment bed elevation, which is set by the balance between sediment erosion, deposition, and rock uplift:

$$\frac{\partial \eta}{\partial t} = \frac{\partial H}{\partial t} = U + D_s - E_s. \tag{41}$$

5  At steady state, $\frac{\partial \eta}{\partial t} = 0$ and $E_s - D_s = U$. Substituting in the equations derived above for sediment erosion and deposition and assuming for simplicity that sediment porosity $\phi$ and the sediment erosion threshold $\omega_{cs}$ are negligible,

$$K_s q^m S^n - V \frac{Q_s}{Q} = U. \tag{42}$$

Applying the steady-state mass conservation relationship $Q_s = UA$, recalling that $Q = rA$, and solving for $S$ gives an expression for steady-state channel slope:

10  $$S = \left[ \frac{UV}{K_s qr} + \frac{U}{K_s q} \right]^{1/n}. \tag{43}$$

If $q = k_q A^m$ as in our test case, the resulting slope-area relationship is then:

$$S = \left[ \frac{UV}{K_s A^m r} + \frac{U}{K_s A^m} \right]^{1/n},$$
(44)

where $k_q$ is subsumed into $K_s$. Eq. (44) nicely distinguishes the contributions of sediment deposition (first term on the right side) and sediment entrainment (second term on the right side) to steady-state channel slope. If effective settling velocity is

negligible, erosion is only limited by the efficiency of sediment entrainment and Eq. (44) gives the detachment-limited steady-state slope (though importantly the bed is still entirely composed of sediment). If entrainment and deposition of sediment are rapid enough that erosion is limited by transport capacity (i.e., the river has enough energy to erode more sediment but the water column is saturated), the system is transport-limited and the left hand term in Eq. (44) dominates in setting the steady-state slope. Note the subtle difference between Eq. (44) and Eq. (18); when $H >> H_*$ and all surface lowering is accomplished by

sediment entrainment, both terms on the right hand side of Eq. (44) reflect erosion of sediment (i.e., $K_s$ is used in both). This occurs because when $H \gg H_*$, change in bedrock elevation over time is not zero as in true complete steady-state, but is equal to the uplift rate. Therefore, for topographic steady state to be achieved, both the transport and detachment terms of Eq. (44) must be accomplished through erosion of sediment. Eq. 44 may be re-written to show that it predicts a standard stream power slope-area relationship that is modified by the ratio of settling velocity to effective runoff:

$$S = \left[ \frac{V}{r} + 1 \right]^{1/n} \left[ \frac{U}{K_s} \right]^{1/n} A^{-m/n}.$$
(45)

We compare the slope-area relationships predicted by Eq. (44) with those extracted from the SPACE model. In order to achieve conditions in which the transport term in Eq. (44) dominates, we set initial soil depth to 100 m everywhere on our test grid so that $H \gg H_*$. See Table 1 for all parameter values.

Figure 7 shows the results of the transport-limited model experiment. The top panel shows the longitudinal profile of

the longest channel, and shows that the SPACE component produces concave-up longitudinal profiles at steady state under transport-limited conditions. The appearance of constant concavity in the longitudinal profile is verified by the constant slope in log-log space of the slope-area data shown in the middle panel of Fig. 7. The bottom panel compares the theoretical slope-area relationship (Eq. (44), gray dashed line in Fig. 2) with data from the model run (black dots). All core nodes from the model domain are included, and all agree well with the theoretical prediction. In addition to matching the analytical prediction for

channel slope, the model also matches the expected steady-state sediment flux relationship, $Q_s = UA$ (Fig. 2, bottom panel). This indicates that the SPACE component is successfully matching expected transport-limited model behavior for both slope and sediment flux at steady state.

### 7.3   Mixed bedrock-alluvial comparison

One major advantage of SPACE over many existing fluvial erosion models is its ability to simultaneously compute the evolution

of an alluvial layer and a bedrock surface. True steady state in the mixed bedrock-alluvial case occurs when the thickness of the alluvial layer $H$ and the bedrock height $R$ are both unchanging in time ($\frac{\partial H}{\partial t} = 0$ and $\frac{\partial R}{\partial t} = 0$). In such a scenario, $E_s = D_s$ and

$U = E_r$. As described in Sect. 4.5, steady-state analytical solutions exist for channel slope, sediment thickness, and sediment flux (here again we use $q = k_q A^m$ with $k_q$ subsumed into $K_s$ and $K_r$, and keep $\phi = 0$ and $F_f = 0$):

$$S = \left[ \frac{UV}{K_s A^m r} + \frac{U}{K_r A^m} \right]^{1/n}, \tag{46}$$

$$H = -H_* \ln \left[ 1 - \frac{V}{\frac{K_s r}{K_r} + V} \right], \tag{47}$$

and

$$Q_s = UA. \tag{48}$$

Running the SPACE component to complete steady state in a case where both sediment entrainment and bedrock erosion contribute to setting channel slope should therefore result in a concave-up channel profile with a sediment layer of constant thickness, and sediment flux equal to the product of the rock uplift rate and drainage area. We use the same tilted plane initial model domain as described in Sect. 7 to test whether the model can replicate the expected behavior in the bedrock-alluvial case. Matching the steady-state analytical solutions requires both erosion of bedrock to generate the concave-up profile, and accumulation of sediment to a constant thickness over the landscape. We begin the numerical experiment with zero sediment thickness at all nodes. Table 1 shows all parameter values used. The driver script used for this model experiment is included in the code guide for this paper.

Figure 8 shows the evolution of the longitudinal bedrock profile and alluvial cover layer in the longest channel over several model timeslices. Beginning from a low-slope tilted plane, the channel incises bedrock and begins to build up a layer of alluvium on the channel bed. As model time progresses, the channel profile increases in concavity and the layer of alluvium thickens. Alluvial thickening progresses from downstream to upstream. By the final timeslice, the alluvial layer has reached its equilibrium value, the channel profile has equilibrated to the imposed uplift rate, and the bedrock surface, alluvium thickness, and topographic surface are all at steady state. Figure 9 shows the final channel profile (top panel) when the topographic surface, sediment thickness, and bedrock height are all at steady state. The topographic surface (top of the sediment layer) is everywhere parallel to the bedrock surface, and the bed sediment layer is 1.25 m thick at every point along the channel profile. Given $V = 5$ m/yr, $K_s = 0.01$ m$^{-1}$, and $K_r = 0.005$ m$^{-1}$, as used in the model, the steady-state sediment thickness of $H = 1.25$ m calculated by the model matches the analytical prediction of Eq. (21). The middle panel compares the theoretical prediction for slope-area scaling given by Eq. (46) (gray dashed line) with the model results (black dots). As in the detachment-limited and transport-limited cases, the model matches the analytical prediction. The bottom panel of Fig. 9 compares the theoretical steady-state relationship between drainage area and sediment flux ($Q_s = UA$) with modeled sediment flux, and shows that the model shows the predicted linear increase in $Q_s$ with drainage area. The ability of the SPACE component to treat both the detachment-limited and transport-limited end members of fluvial systems as well as the mixed bedrock-alluvial case confirms that the model equations are being solved correctly, and importantly that our use of stabilizing, local analytical solutions does

not compromise the ability of the model to replicate expected behavior. Below, we show how the SPACE component may be efficiently coupled with other surface processes models in the Landlab modeling framework to provide insight into landscape evolution.

## 8 Application to landscape evolution modeling: Coupling SPACE with hillslope diffusion to model topographic growth and decay

One frequent application of landscape evolution modeling is the exploration of landscape response to tectonic perturbations. Understanding the growth and decay of topography has significant implications for interpretation of the stratigraphic record, which is composed of sediment that is detached and transported from upland landscapes. In this section we show how the SPACE component can be coupled with a hillslope diffusion model in the Landlab modeling toolkit to simulate landscape response to changing rock uplift rates. In addition to computing topographic change that incorporates both sediment and bedrock surface evolution, we show the capability of the SPACE component to provide information about sediment fluxes delivered from the model catchment over time.

### 8.1 Model setup

We use a Landlab raster model grid composed of 2500 nodes (50 x 50 grid). We use a node spacing ($\mathrm{d}x$) of 100 m, resulting in a 25 km$^2$ area grid. As with the model verification experiments described above, we close all model domain boundaries except for a single open outlet in the lower-left (southwest) corner. The use of a single outlet means that the entire model domain will be a single watershed draining to the outlet, and the sediment flux leaving the outlet is the integrated sediment flux from the entire basin. Our landscape initial condition is a plane tilted slightly (initial regional slope of $\approx 1.4 \times 10^{-5}$) towards the basin outlet. The tilted plane has initial, random sub-millimeter scale surface roughness to initiate the formation of drainage pathways.

For simplicity we use the simple stream power form of the SPACE model ($m = 0.5$, $n = 1$, $q = k_q A^m$) and keep $\phi = 0$ and $F_f = 0$. We do not incorporate sediment entrainment and bedrock erosion thresholds. See Table 1 for all SPACE parameter values used in this example. We couple this parameterization of the SPACE component with Landlab's linear diffusion component, which computes the topographic-gradient-driven movement of mass at every node by the equation

$$\frac{\partial \eta}{\partial t} = \kappa \frac{\partial^2 \eta}{\partial x^2} \tag{49}$$

where $\kappa$ is a diffusivity in units of [L$^2$/T] (we use $\kappa = 0.005$ m$^2$/yr). Programmatically, we incorporate Eq. (49) simply by running Landlab's linear diffusion component immediately after running the SPACE component in each model timestep. One simplification made by simple linear diffusion models is that the entire landscape is made of the same material (i.e., no distinction between rock and soil). To realistically couple linear diffusion with our fluvial erosion model that explicitly separates the dynamics of sediment and bedrock, we assume that any material diffused from one model node onto another is sediment.

Such an assumption is realistic given the purpose and limitations of the linear diffusion model, which has been shown to apply primarily to soil-mantled hillslopes.

We ran our coupled model with a 1 year timestep for 300,000 years. Because our goal was to use SPACE and linear diffusion to explore topographic growth and decay, we used a rock uplift rate $U$ relative to baselevel that was unsteady in time. We simulated a "pulse" of rapid rock uplift preceded and followed by periods of slower rock uplift. $U$ was set to 0.0001 m/yr for the first 100 kyr, increased to 0.0005 m/yr for the second 100 kyr, and returned to 0.0001 m/yr for the final 100 kyr. Two variables of broad interest that generally vary in response to changing rock uplift rates are topographic relief and total sediment flux out of the model catchment. We predict under such a scenario that topographic relief and sediment flux will increase as surface erosion responds to rock uplift for the first 200 kyr, and that relief and sediment flux will reach their maximum when the rock uplift rate is at its highest value. We then expect decays in relief and sediment flux as the high points in the landscape erode and topographic gradients shrink in response to a return to the lower rock uplift rate. We recorded the topographic elevation at all model nodes, along with relief and sediment flux from the domain, at 1 kyr intervals.

## 8.2   Results of coupled model experiment

Figure 10 shows several timeslices of model topography showing the response to imposed rock uplift, and Fig. 11 shows sediment depth over the model domain at the same timeslices. Over the first 100 kyr, bedrock incision into the initially non-alluviated domain dominates adjustment to rock uplift. However, because the rock uplift rate is low and topographic gradients are being lowered by fluvial erosion and hillslope diffusion, the landscape has $< 10$ m of relief after 100 kyr. Sediment is first produced on the hillslopes (the products of diffusion are considered to be sediment) but not stored in the channels for the first 50 kyr as the channels incise into bedrock to accommodate the onset of rock uplift. By 100 kyr, the landscape is nearly equilibrated to the rock uplift rate, and alluviation has occurred in the channels as sediment thickness approaches its equilibrium value everywhere. Between 100 and 200 kyr, during which the rock uplift rate is five times its initial value, the network progressively develops higher and higher relief (up to $\approx 50$ m) until the rock uplift rate is reduced again at 200 kyr. During this period, the channels initially strip their alluvial cover and incise bedrock to match the increased rock uplift rate. However, as in the first 100 kyr, as the landscape begins to equilibrate to the new rock uplift rate by 200 kyr, alluvium thickness again increases towards its equilibrium condition. Once the rock uplift rate is reduced, diffusion of material from the hillslopes into the channels results in sediment mantling of the channels and significant reduction of the rate of bedrock incision. By 300 kyr, the lowering of high points by diffusion, together with the inability of low-gradient, sediment-mantled rivers to effectively incise bedrock, has resulted in reductions in landscape relief such that the landscape at 300 kyr nearly mirror the 100 kyr timeslice ($\approx 10$ m of relief). At this point, the reduction in topographic gradients has reduced sediment delivery to the channels, which begin to strip their alluvial cover down to its equilibrium thickness.

The patterns observed in Fig. 10 and Fig. 11 are quantified in Fig. 12. Relief and sediment flux out of the model domain increase initially as the landscape is adjusting, through bedrock incision, from its initial condition to become equilibrated with the imposed rock uplift rate. The rates of increase in both relief and sediment flux increase substantially as the rock uplift rate quintuples at 100 kyr, and while neither reaches its equilibrium value (i.e., steady in time), both relief and sediment

flux begin to asymptote towards those equilibrium values. At 200 kyr when the rock uplift rate is reduced to its initial value, diffusion rapidly reduces topographic gradients and alluviates channels, and channels become less erosive. Thus both relief and sediment flux decline over the final 100 kyr of the experiment, again approaching, but not fully reaching, their steady-state values. The simple experiment performed here shows that the SPACE component may be easily coupled with other models of Earth surface processes in Landlab to explore any number of questions relating to landscape evolution. While 1-D models of river longitudinal profile evolution have proliferated widely over the past decades, models that explicitly incorporate sediment morphodynamics and bedrock erosion simultaneously (e.g., Lague, 2010; Nelson and Seminara, 2011; Inoue et al., 2014, 2016, 2017; Zhang et al., 2015) are rare. Those that act over 2-D landscapes and are easily coupled with other models are even rarer. The SPACE algorithm and component fill an important gap in the quantitative geomorphologists's toolkit, and it is our hope that future users will apply it, potentially in conjunction with other Landlab components, to solve diverse geomorphological problems.

## 9 Discussion

In this section, we first explore the possibilities for testing the SPACE model against real and experimental landscapes. We then summarize the limitations of the model, and compare SPACE with existing models of bedrock-alluvial channel evolution.

### 9.1 Potential for model validation against real landscapes

Models of river erosion are notoriously difficult to test against field data. Uncertainties in initial and boundary conditions, as well as spatial and temporal heterogeneity in rock type and sediment size, introduce substantial complexity into model verification exercises. Nevertheless, studies exploiting natural experiments, or landscapes where initial conditions, boundary conditions, and/or parameter values are particularly well-constrained (Tucker, 2009), have met with some success in validating models (e.g., Stock and Montgomery, 1999; Tomkin et al., 2003; van der Beek and Bishop, 2003; Valla et al., 2010; Hobley et al., 2011). The SPACE model makes predictions for steady-state channel slope and sediment thickness that could be validated by field or experimental studies. In addition, the co-evolution of the bedrock surface and sediment layer thickness could be used to validate the SPACE model in transient cases.

Previous studies have addressed the validation of the detachment-limited and transport-limited end-members of fluvial erosion models (e.g., Stock and Montgomery, 1999; Tomkin et al., 2003; van der Beek and Bishop, 2003; Valla et al., 2010; Hobley et al., 2011), so we focus here on the potential for evaluating the predictions of SPACE for the mixed bedrock-alluvial case. The SPACE model predicts that both sediment entrainment and bedrock erosion will contribute to setting the steady-state slope-area relationship (Eq. 46), and that the relative importance of the two components will be set by the ratio $\frac{V}{r}$ and the relative erodibility of sediment and bedrock. The steady-state sediment layer thickness (Eq. 47) is also governed by $\frac{V}{r}$ and the ratio of erodibilities. As $\frac{V}{r}$ or $\frac{K_r}{K_s}$ increases, the channel slope is dominated by sediment entrainment dynamics and the steady-state alluvial layer thickness increases. If the SPACE model is valid for natural settings, it should be possible to find (or create in the laboratory) steady-state channels with an alluvial layer with constant thickness along the channel. Further,

across a gradient of rock and/or sediment erodibilities (e.g., a sequence of changing metamorphic grade or a spatial gradient in grain sizes), both the steady-state channel slope and steady-state alluvial thickness should show predictable changes as a function of $\frac{K_r}{K_s}$. Finally, an intriguing prediction of SPACE is that the steady-state alluvial thickness is independent of the rock uplift rate while the slope-area relationship is not. Field investigations across a range of rock uplift rates with consistent lithology, climate, and sediment properties should show constant steady-state sediment thickness imposed on rock-uplift-dependent slope-area scaling.

In the transient case, the relationship between the longitudinal profile of the bed sediment surface and that of the bedrock may be useful for validating SPACE model predictions. For example, Fig. 8 shows that for an uplifting landscape with zero initial sediment thickness, SPACE predicts bottom-to-top alluviation of the channel profile. In this case, the sediment surface does not reflect the steepened reach commonly associated with the propagation of transient signals up a river profile, while the bedrock beneath does. The prediction of SPACE is therefore that in a transient river profile with some amount of bed sediment, the concavity of the sediment surface is not expected to match that of the bedrock surface. The difference in concavity between the sediment surface and the bedrock surface should then decline as channels approach steady state, a prediction that is testable in a landscape where channels exist in different stages of transient adjustment. It is important to remember that the sediment thickness predicted by SPACE is a spatial average within a model cell. Further, using realistic (i.e., time-varying) flow distributions to force the model would result in temporal variability in sediment thickness (Lague et al., 2005; Lague, 2010), complicating the interpretation of sediment thickness values from a given field campaign. While testing the steady-state predictions of SPACE is likely feasible in well-constrained landscapes, the transient dynamics may be best explored in a laboratory setting.

## 9.2 Limitations of the SPACE model

The SPACE model is intended to provide a simple, extensible, easy-to-use tool to expand the set of questions that can be addressed with numerical models of river channel evolution. We have consolidated recent advances in the treatment of sediment erosion and deposition (Davy and Lague, 2009) as well as simultaneous evolution of sediment and bedrock layers (Lague, 2010; Zhang et al., 2015) into a single model that has the additional advantages of being effective for modeling over large grids on landscape evolution timescales, easily accessible, and easily coupled to other surface processes models in the Landlab modeling toolkit. As such, our model encompasses some of the same limitations as the previous work from which it is derived, including a lack of dynamic channel width and a lack of treatment of the tools effect.

To allow efficient solution of the SPACE model over large model grids and timescales, we do not incorporate the hydro-dynamic calculations (i.e., explicit computation of stresses on the bed and banks) required to allow channel width to evolve freely (e.g., Kean and Smith, 2004; Stark, 2006; Wobus et al., 2006; Davy and Lague, 2009; Turowski et al, 2009; Lague, 2010; Nelson and Seminara, 2011; Coulthard et al., 2013). Instead, we employ a common parameterization for channel width as a function of drainage area or discharge. In the examples in this paper we used a width scaling of $w \approx Q^{0.5}$, which results in $q \approx Q^{0.5}$ or $q \approx A^{0.5}$ if drainage area is linearly related to volumetric water discharge. The width scaling exponent $m$, which we held equal to 0.5 in the examples discussed here, is a user-defined parameter. The use of an empirical width scaling means

that 1) any given point along the channel occupies a single grid cell regardless of whether width is less than, equal to, or greater than grid cell size, and 2) SPACE does not capture temporal channel width variations in response to rock uplift, sediment flux, and discharge changes. In some cases, channel width may respond more significantly than channel slope to such forcings (e.g., Amos and Burbank, 2007; Turowski et al, 2009). While our use of a downstream width relationship enables us to test the SPACE model against known analytical solutions and eliminates the need for hydrodynamic calculations, the future addition of dynamic width would allow exploration of the relative importance of slope and width adjustment.

The SPACE model as presented here follows previous models in treating reduction of bedrock erosion by sediment cover (e.g., Beaumont et al., 1992; Lague, 2010; Shobe et al., 2016), but does not include the enhancement of bedrock erosion by the presence of mobile bedload tools (e.g., Sklar and Dietrich, 2004; Gasparini et al., 2006; Turowski et al., 2007; Chatanantavet and Parker, 2009; Zhang et al., 2015). Such a simplification keeps expected model behavior conceptually simple and allows comparison of model results to known analytical solutions for the purposes of model validation. However, there is substantial field evidence indicating that mobilized bedload can be an important erosive agent, both by detaching bedrock particles and extending fractures through macroabrasion (Hancock et al., 1998; Cook et al., 2013; Beer et al., 2016). The major effect of excluding this effect from the SPACE model is that bedrock erosion is underpredicted at low to moderate sediment fluxes, where there is enough bedload to frequently abrade the bed, but not enough to form a deep layer of alluvial cover. Adding the tools effect to the SPACE model could be accomplished by changing the form of the bedrock erosion function to increase with increasing $Q_s$.

### 9.3   Comparison to previous channel evolution models

The SPACE model borrows and combines concepts from models for sediment-flux-dependent bedrock incision, bed cover evolution, sediment transport, and prior models of mixed bedrock-alluvial channels. Here we briefly highlight the similarities and differences between our model and previous models. SPACE relaxes the traditional assumptions governing detachment-limited and transport-limited erosion models by incorporating both mass conservation of sediment and an incision rule for bedrock. Our model, like other recent models for bedrock-alluvial channels, explicitly incorporates sediment transport to move beyond the assumption that bed exposure depends on the ratio of sediment flux to transport capacity used in sediment-flux-dependent bedrock incision models (e.g., Sklar and Dietrich, 2004; Gasparini et al., 2006; Turowski et al., 2007; Chatanantavet and Parker, 2009). In doing this, our model follows a substantial number of previous contributions. Specifically, Hodge and Hoey (2012), Nelson and Seminara (2012), and Turowski and Hodge (2017) used explicit treatments of sediment morphodynamics to explore spatial and temporal changes in bed cover on a non-erodible bedrock bed in response to different forcings. Turowski (2009) used a stochastic sediment erosion and deposition framework in conjunction with the saltation-abrasion model for bedrock erosion to explore both cover variation and its influence on bedrock erosion rates. Their models are formulated at the reach scale and focus on cover dynamics rather than landscape evolution. Unlike SPACE however, their models allow different forms for the dependence of exposed bed fraction on sediment transport to arise dynamically, while we use a simple exponential decline in bed exposure with increasing sediment thickness. Nelson and Seminara (2011) coupled sediment transport saltation-abrasion rules in a model of channel cross-section evolution. Our model assumes a planar channel bed and parameterized channel width

for application to landscape evolution problems, and does not dynamically evolve channel cross-section shape as does the model by Nelson and Seminara (2011). Sediment transport in the SPACE model is computed with the erosion-deposition framework, using an approach almost exactly following Davy and Lague (2009). The major difference between SPACE and the models of Davy and Lague (2009) and Carretier et al. (2016) is that SPACE applies the erosion-deposition framework to the simultaneous erosion of sediment and bedrock. This allows the model to transition not only between transport-limited and detachment-limited behavior, as in those models, but also between fully alluviated, mixed bedrock-alluvial, and pure bedrock states. This is also the key difference between SPACE and the Fowler et al. (2007) model, which used Exner-based sediment conservation and bed abrasion under an assumption of infinite sediment supply.

Inoue et al. (2014, 2016, 2017) formulated three insightful models for the evolution of mixed bedrock-alluvial channels. There are several areas in which these models incorporate more physical realism in fluvial erosion processes than does SPACE. For example, Inoue et al. (2014) incorporated the roughness of both the bedrock surface and the alluvial layer; their model is therefore able to treat both the case where the alluvial layer is rougher than the bedrock bed and vice versa. SPACE does not currently have the ability to alter flow resistance based on the relative exposure of sediment and bedrock. While more advanced representations of the roughness of both bedrock and alluvium are possible, they are not currently incorporated in our model. However, the erosion-deposition treatment of sediment dynamics in SPACE means that it can move beyond the constant sediment supply assumption of Inoue et al. (2014). The models of Inoue et al. (2016) and Inoue et al. (2017) contain substantially more reach-scale complexity than SPACE in that the flow and sediment transport equations are solved in 2-D, resulting in their model being able to resolve changes to alluvial thickness and bedrock elevation in the downstream and cross-stream directions. Inoue et al. (2017) also added a simple parameterization for bank erosion. The SPACE model treats bedrock elevation and alluvial thickness as cell-averaged quantities, and channel width is empirically parameterized and therefore may at any point be smaller than, equal to, or greater than a cell width. As such, SPACE resolves downstream changes in sediment thickness and bedrock elevation, but not cross-stream changes. On the other hand, the simplicity of treating the problem in 1-D allows SPACE to be applied to orogen-scale grids over landscape evolution timescales, whereas the models of Inoue et al. (2014, 2016, 2017) have primarily been applied at the reach to kilometer scale.

SPACE is most similar to, and most based upon, the models of Lague (2010) and Zhang et al. (2015), both of which model the full transition between bedrock, bedrock-alluvial, and fully alluvial channels over full channel profile scales. All three models contain similar treatments of the progressive exposure of bedrock with thinning of the alluvial layer. Lague (2010) compared alluvial layer thickness to the median grain size, Zhang et al. (2015) compared alluvial thickness to the macro-roughness of the bedrock surface, and SPACE similarly computes bedrock exposure based on the ratio of alluvial thickness to a bedrock roughness length scale. The three models differ primarily in that SPACE uses the erosion-deposition framework of Davy and Lague (2009) to explicitly calculate sediment transport morphodynamics, while Lague (2010) and Zhang et al. (2015) use Exner-based sediment conservation approaches. The other differences among the three models are driven by which processes are included and which are neglected. The model of Lague (2010) incorporates dynamic channel width by partitioning shear stresses between the bed and banks, while both the model of Zhang et al. (2015) and SPACE do not incorporate dynamic channel width, but use empirical width scaling parameterizations. Zhang et al. (2015) use a saltation-abrasion rule for bedrock

incision in their model such that they capture both the tools and cover effects. Both the model of Lague (2010) and the SPACE model incorporate bed cover as a function of increasing alluvial layer thickness, but do not include a dependence of bedrock erosion on sediment flux (the tools effect). Both dynamic width and the tools effect could be incorporated in future versions of the SPACE model.

## 10 Conclusions

We have developed and presented a new model for sediment transport and river incision into bedrock. The SPACE model takes inspiration from sediment-flux-dependent bedrock incision models (e.g., Sklar and Dietrich, 2004; Gasparini et al., 2006; Turowski et al., 2007), models of alluvial cover on bedrock beds (e.g., Turowski, 2009; Hodge and Hoey, 2012; Nelson and Seminara, 2011, 2012; Turowski and Hodge, 2017), erosion-deposition sediment transport models (e.g., Beaumont et al., 1992; Davy and Lague, 2009), and existing models of bedrock-alluvial river evolution (e.g., Lague, 2010; Inoue et al., 2014; Zhang et al., 2015; Inoue et al., 2016, 2017). SPACE incorporates explicit sediment erosion and deposition and simultaneously evolves a sediment layer and a bedrock bed, with the necessary simplifications (e.g., parameterized width) to make solutions tractable over landscape evolution spatial and temporal scales.

We developed steady-state analytical solutions for channel slope, sediment thickness, and sediment flux in Sect. 4.5 based on the model governing equations. We then showed three experiments demonstrating that the numerical implementation of SPACE, which uses local analytical solutions for stable calculation of sediment flux and sediment thickness, matches the analytical predictions at steady state in the detachment-limited, transport-limited, and mixed bedrock-alluvial cases.

The foremost advantage of SPACE over other similar models is its ease of use for modeling landscape evolution in two dimensions. The SPACE 1.0 component is implemented in 2-D as part of the freely available Landlab modeling toolkit, and may be readily coupled to other models of Earth surface processes. We showed in an example application how the SPACE component may be coupled to a linear diffusion hillslope evolution model to investigate the growth and decay of topography in response to temporally variable rock uplift. The ability of SPACE to separately evolve bedrock topography and sediment thickness makes it well-suited to a suite of possible applications that are out of reach for simpler, single-substrate models. For example, the SPACE component would be effective for modeling the depositional filling of flexural depressions, downthrown fault blocks, and landslide-dammed rivers. In situations where temporal variability in sediment flux is a variable of interest, the ability of SPACE to store sediment in the form of an alluvial layer could yield more realistic results than simple detachment- or transport-limited models. Finally, the SPACE modeling framework is flexible in that the stream-power based entrainment and erosion equations (Eq. (7 and 8)) may be replaced with other formulations better tailored to individual model applications.

The SPACE 1.0 component enables 2-D calculation of sediment transport, bedrock erosion, and landscape evolution within the Landlab modeling toolkit. The model's ability to simultaneously transport sediment and erode bedrock opens up a wide variety of potential landscape evolution applications beyond the limits of simpler models. SPACE may be easily coupled to other models in Landlab to address novel questions in geomorphology.

## 11 Code availability

The SPACE 1.0 Landlab component as well as all other Landlab components used in this paper are part of Landlab version 1.0.2. Source code for the Landlab project is housed on GitHub: http://github.com/landlab/landlab. Documentation, installation instructions, and software dependencies for the entire Landlab project can be found at http://landlab.github.io/. A detailed user manual with an accompanying Jupyter notebook and a driver script for the mixed bedrock-alluvial example illustrated in this paper can be found at https://github.com/cmshobe/pub_shobe_etal_GMD (Shobe, 2017, GitHub Repository). The Landlab project is tested on recent-generation Mac, Linux, and Windows platforms using Python versions 2.7, 3.4, and 3.5. The Landlab modeling framework is distributed under a MIT open-source license.

*Author contributions.* GET developed the algorithm with help from CMS. CMS implemented the algorithm, wrote the Landlab component with help from KRB, verified and evaluated the model solutions, and wrote the paper with contributions from GET and KRB.

*Competing interests.* The authors declare that they have no conflicts of interest.

*Acknowledgements.* This work was supported by National Science Foundation grants ACI-1147454 (PI: Gregory E. Tucker) and ACI-1450409 (PI: Gregory E. Tucker), a National Defense Science and Engineering Graduate fellowship (to CMS), and a University of Colorado Chancellor's fellowship (to CMS). Thanks to Jordan Adams, Alexandra Carriere, Rachel Glade, Harrison Gray, Sarah Harbert, Aaron Hurst, Kelly Kochanski, Caroline Le Bouteiller, Matt Rossi, and Ronald van Balen for helpful discussions. We thank Fiona Clubb, Jens Turowski, and Associate Editor Jeffrey Neal for insightful and constructive reviews.

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

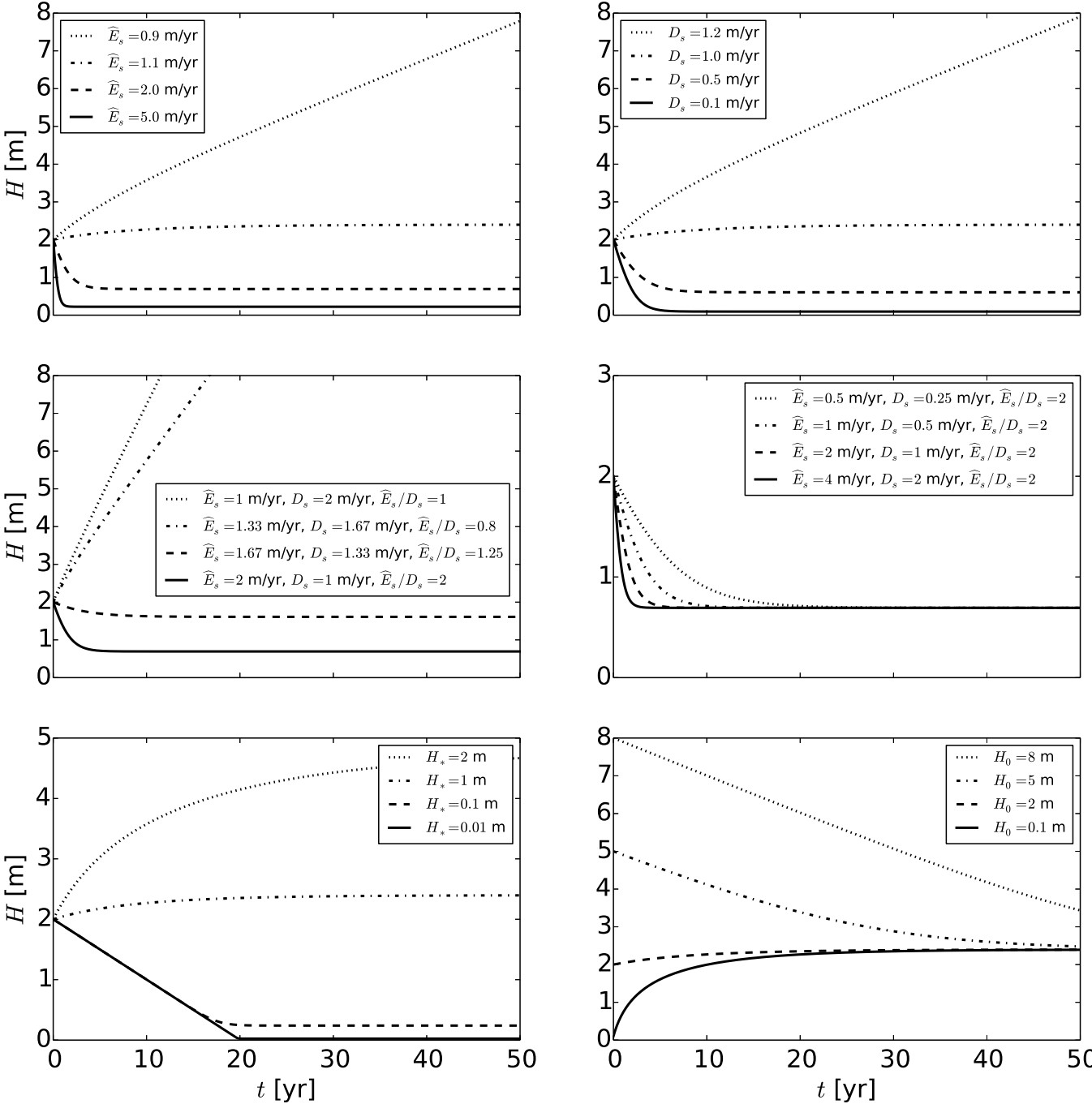

**Figure 4.** Bed sediment thickness $H$ as a function of time as calculated by our local analytical solution (general case for $D_s/(1-\phi) \neq \widehat{E}_s$). In each panel, a single parameter (or set of parameters in the case of $\widehat{E}_s/D_s$) in Eq. (32) is varied while all others are held constant. $H_0$, the initial sediment thickness, sets the initial value of the function. The value of $H$ approached over long timescales is set by competition between the rates of sediment erosion and deposition, where higher sediment erosion rates drive bed sediment thickness down (upper left panel) and higher deposition rates result in greater bed sediment thickness (upper right panel). Except in cases where $D_s > E_s$, our local analytical solution converges on a constant value as $t \to \infty$. When $D_s > E_s$, the solution converges to a simple linear extrapolation of the deposition rate over time. Note that the adjustment time changes with different values of $\widehat{E}_s$ and $D_s$ even when $\widehat{E}_s/D_s$ remains constant. Parameter values (except where changed in the six panels) are: $H_* = 1.0$ m, $D_s = 1.0$ m/yr, $\widehat{E}_s = 1.1$ m/yr (resulting in $E_s = 0.95$ m/yr

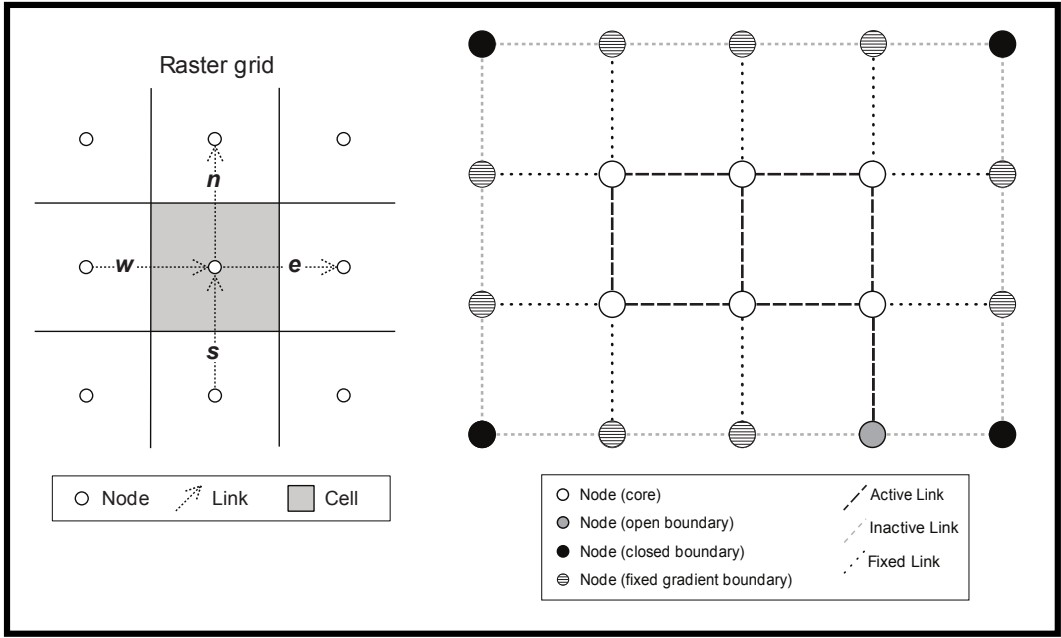

**Figure 5.** Left: the Landlab structured raster model grid, with definitions for major grid elements. State variables such as sediment depth and sediment flux are stored at grid nodes. While gradients such as topographic slope are calculated along links, the slope value representing the steepest descent from a node to its flow receiving neighbor is stored on the node itself. Link direction is topological; the direction of fluxes is set by gradients along links. Figure reproduced from Fig. 3 and 4 in Adams et al. (2017).

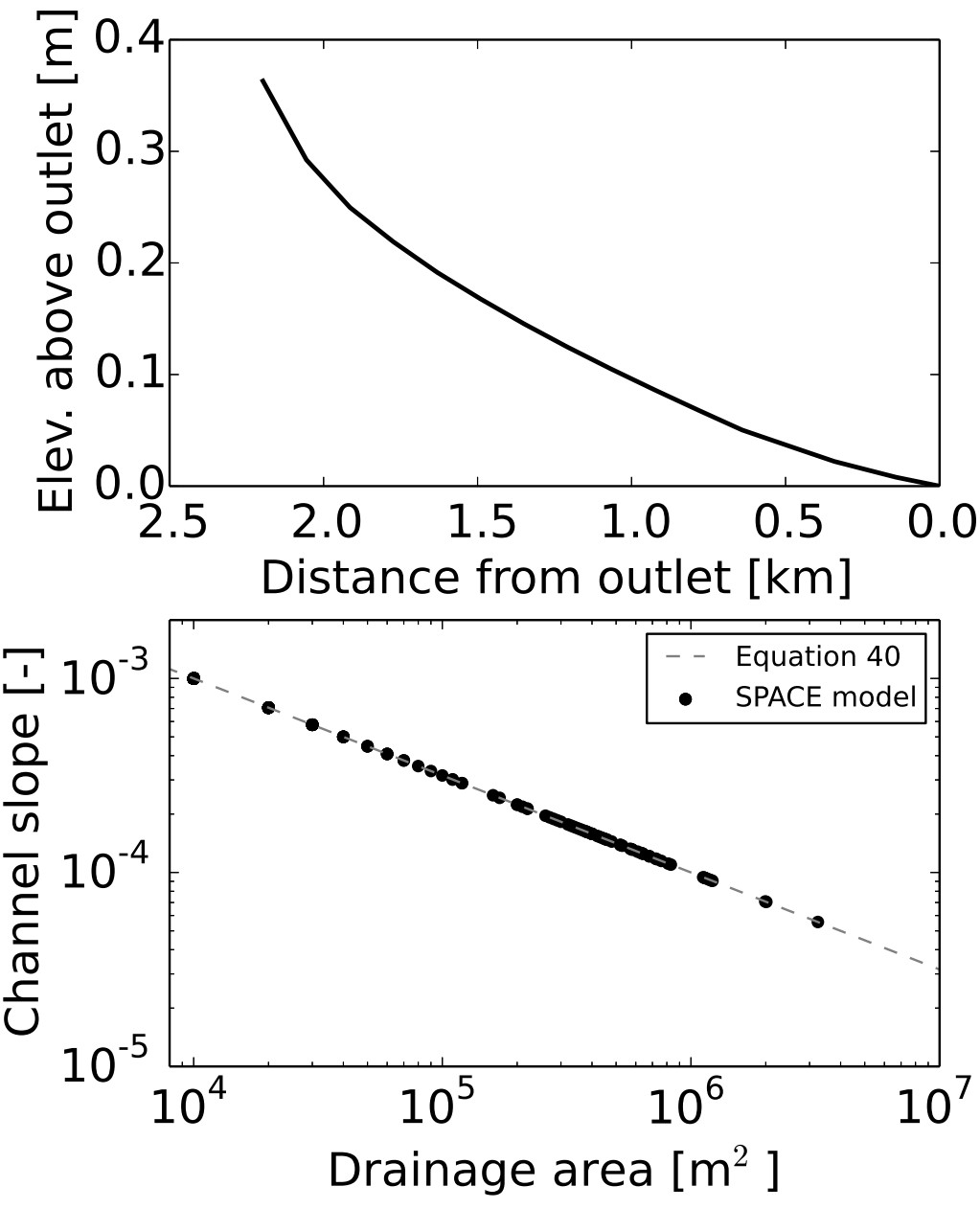

**Figure 6.** Top: longitudinal profile of the longest channel in the model domain under detachment-limited conditions, showing that the channel is in equilibrium with the imposed baselevel fall, and that the SPACE component yields concave-up longitudinal profiles at steady state. Bottom: comparison between the SPACE component and Eq. (40) (steady-state slope-area relationship under detachment-limited conditions). The numerical implementation of the SPACE component successfully replicates the predicted power-law slope-area relationship.

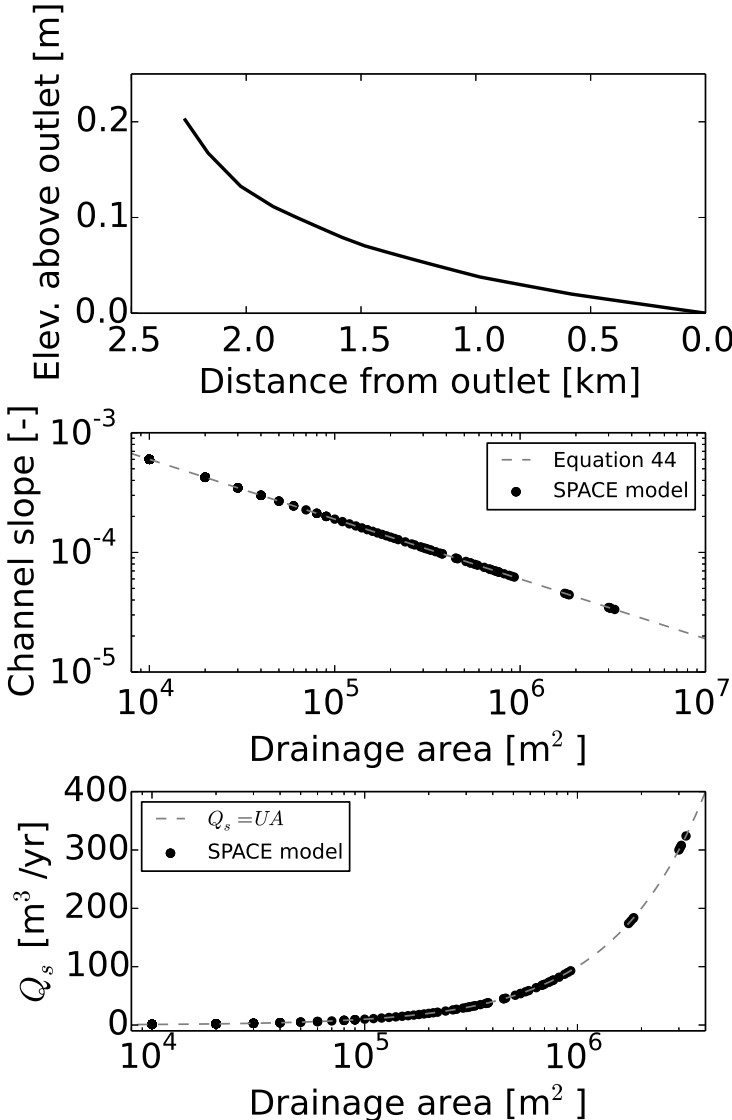

**Figure 7.** Top: longitudinal profile of the longest channel in the model domain under transport-limited conditions, showing that the channel is in equilibrium with the imposed baselevel fall, and that the SPACE component yields concave-up longitudinal profiles at steady state. Middle: comparison between the SPACE component and Eq. (44) (steady-state slope-area relationship under transport-limited conditions). The numerical implementation of the SPACE component successfully replicates the predicted power-law slope-area relationship. Bottom: Sediment flux $Q_s$ as a function of drainage area. The model matches the predicted linear relationship $Q_s = UA$.

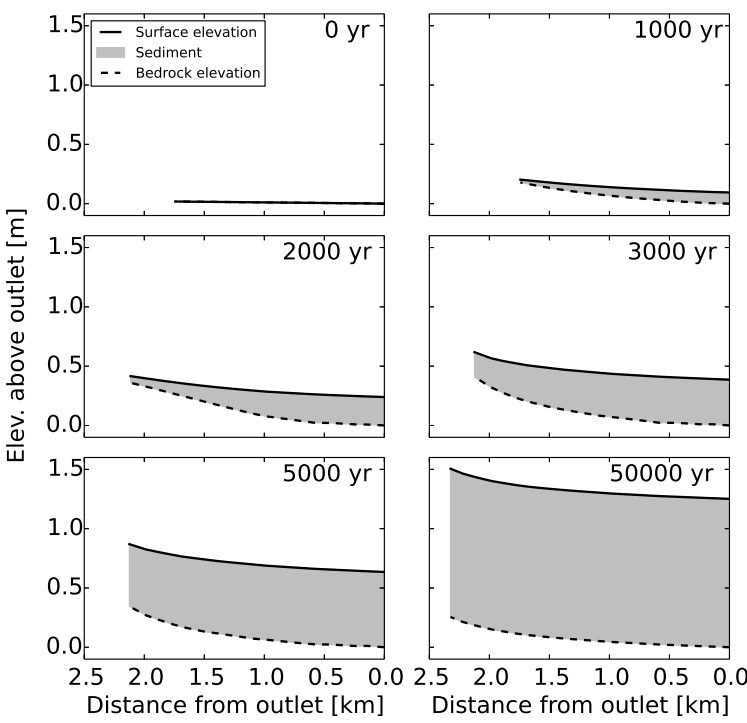

**Figure 8.** Time series of longitudinal profile evolution for the longest channel in the test model domain as the domain is uplifted relative to baselevel. Profile distance lengthens over time as the original tilted ramp is incised; horizontal scale on all plots is the same. Initially (0 yr), the channel topographic surface is effectively flat with zero sediment thickness. By 1000 yr, a slightly concave-up bedrock profile has developed, with a thin, downstream-thickening layer of bed sediment resulting in a surface profile that is less concave-up than the bedrock profile. Over the following three timeslices, continued rock uplift relative to baselevel causes increased concavity in the bedrock profile, as well as continued thickening of the sediment layer. The sediment layer thickness in a downstream to upstream progression. By 50,000 yr, the sediment layer has uniform thickness, resulting in a surface profile of equal concavity to the bedrock profile, and the alluvial layer thickness, topographic surface elevation, and bedrock surface elevation are all equilibrated to the imposed rock uplift rate and are therefore unchanging in time. Vertical exaggeration $\approx 1100\times$.

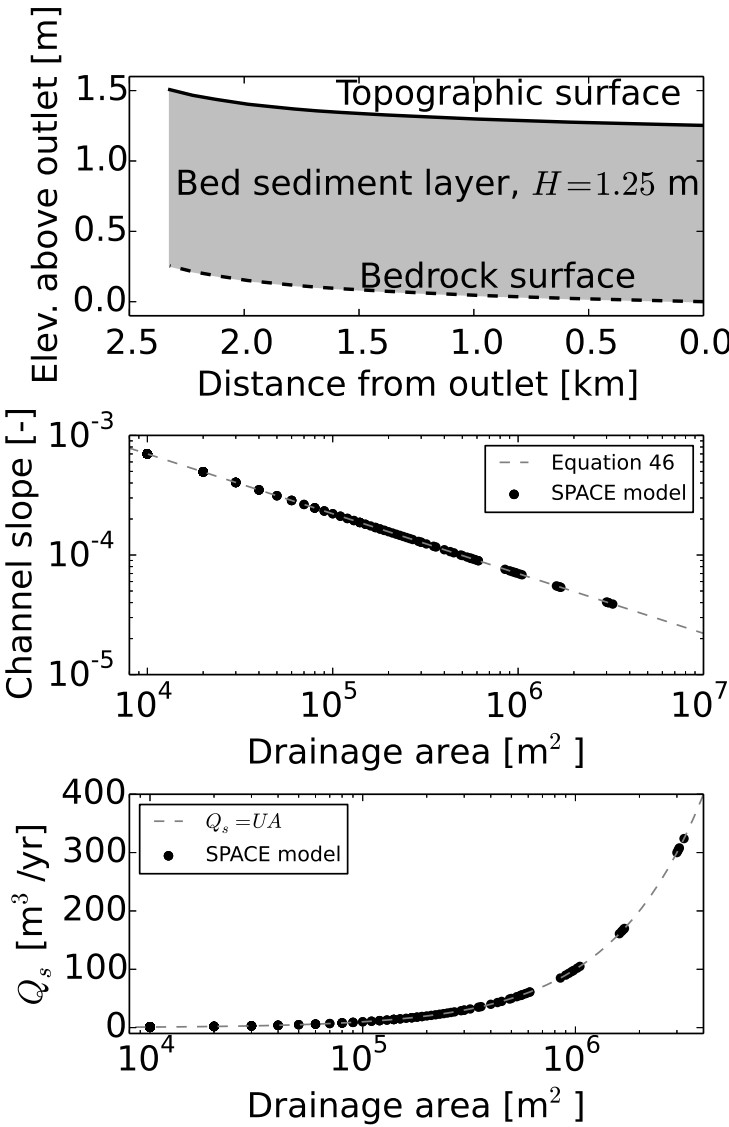

**Figure 9.** Top: longitudinal profile of the longest channel in the model domain after 200 kyr of model time during which the channel has evolved from an initial condition of zero sediment thickness to a constant, steady sediment thickness. Both the topographic surface (top of the sediment layer) and the bedrock surface are in equilibrium with the imposed baselevel fall, showing that the SPACE component yields parallel, concave-up longitudinal profiles in sediment and bedrock at steady state. The sediment thickness $H$ matches the predicted sediment thickness for the parameters used in the model run. Middle: comparison between the SPACE component and Eq. (46) (steady-state slope-area relationship under bedrock-alluvial conditions). The numerical implementation of the SPACE component successfully replicates the predicted power-law slope-area relationship. Bottom: Sediment flux $Q_s$ as a function of drainage area. The model matches the predicted linear relationship $Q_s = UA$.

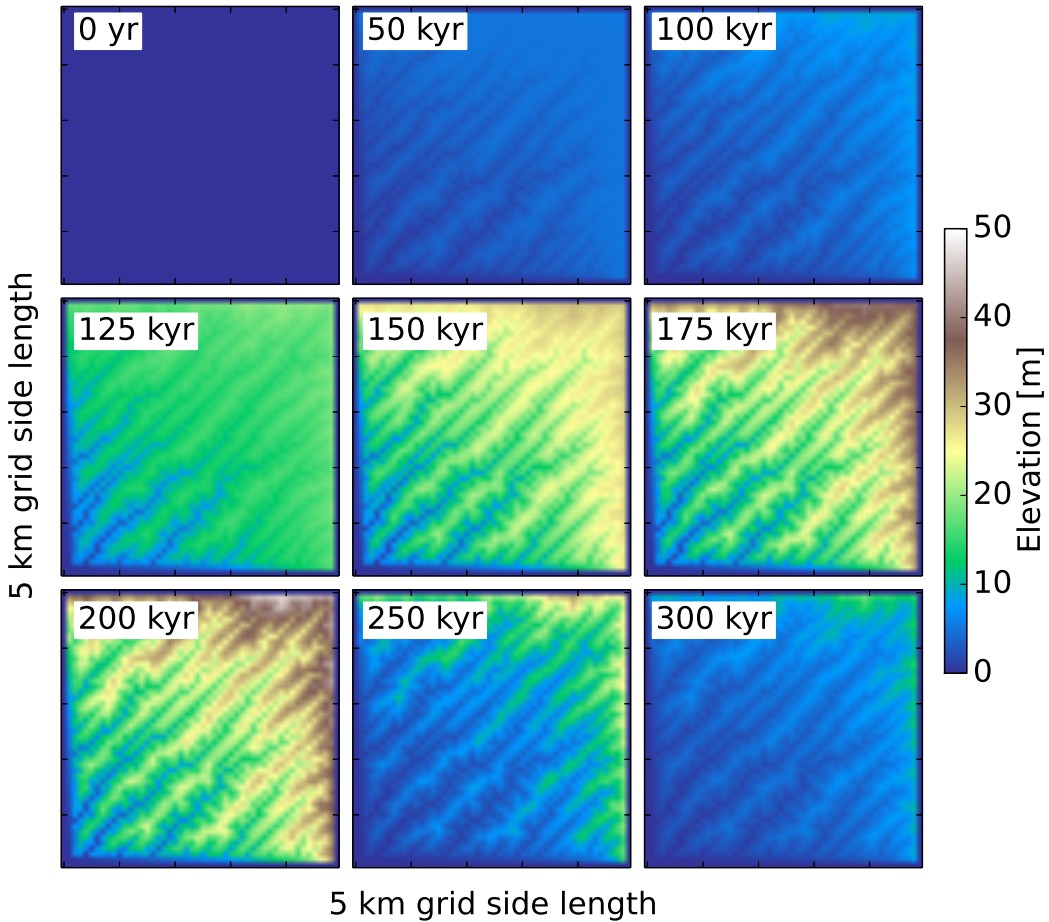

**Figure 10.** Snapshots of topography at nine times throughout the coupled SPACE and linear diffusion model experiment simulating the growth and decay of topography in response to changing rock uplift rates. The upper-left panel represents the model initial condition, a plane with initial micro-scale roughness slightly tilted towards the basin outlet. The outlet is located at the lower-left (southwest) corner of the model domain. The rock uplift rate is 0.0001 m/yr for the first 100 kyr, increases to 0.0005 m/yr for 100–200 kyr, then declines again to 0.0001 m/yr for 200-300 kyr. The model shows the growth of relief in response to rock uplift, with relief increasing slowly for the first 100 kyr and then more quickly for 100–200 kyr. After 200 kyr when the uplift rate declines to its initial value, relief declines as erosion of the high points on the landscape, driven by both fluvial erosion and linear diffusion, outpaces rock uplift.

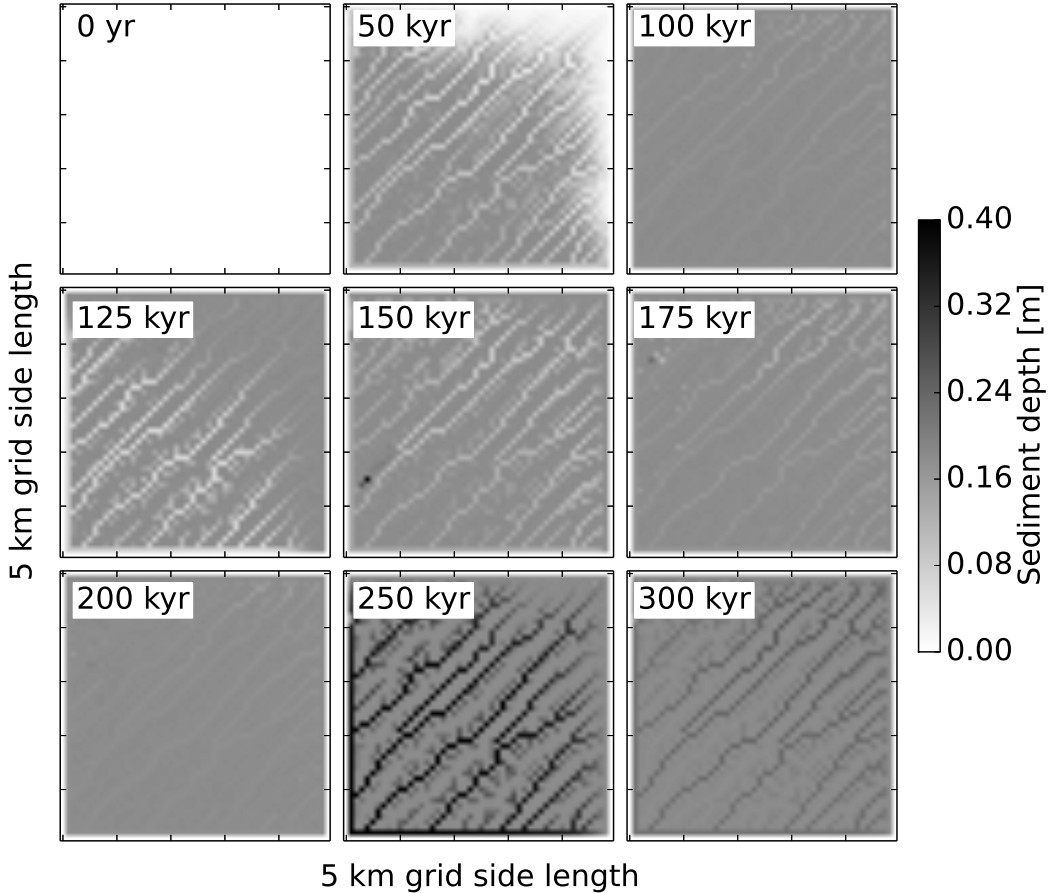

**Figure 11.** Snapshots of sediment thickness at nine times throughout the coupled SPACE and linear diffusion model experiment simulating the growth and decay of topography in response to changing rock uplift rates. The upper-left panel represents the model initial condition, a plane with initial micro-scale roughness slightly tilted towards the basin outlet. The outlet is located at the lower-left (southwest) corner of the model domain. The rock uplift rate is 0.0001 m/yr for the first 100 kyr, increases to 0.0005 m/yr for 100–200 kyr, then declines again to 0.0001 m/yr for 200-300 kyr. During the first 100 kyr, sediment depth increases on the hillslopes (because diffused material is considered sediment) but is absent in the channels as the channels incise into bedrock. By 100 kyr, the channels hold more sediment as incision and sediment thickness equilibrate to the rock uplift rate. As the rock uplift rate increases between 100 and 200 kyr, the same pattern occurs where sediment is evacuated from the channels during the initial response to rock uplift. The channels then re-alluviate as the landscape approaches equilibrium at 200 kyr. After 200 ky, the decline in rock uplift causes initial alluviation in the channels as diffusion into the channels outpaces sediment erosion. Finally, sediment thickness declines in the channels as the pace of diffusion slows in response to relief reduction.

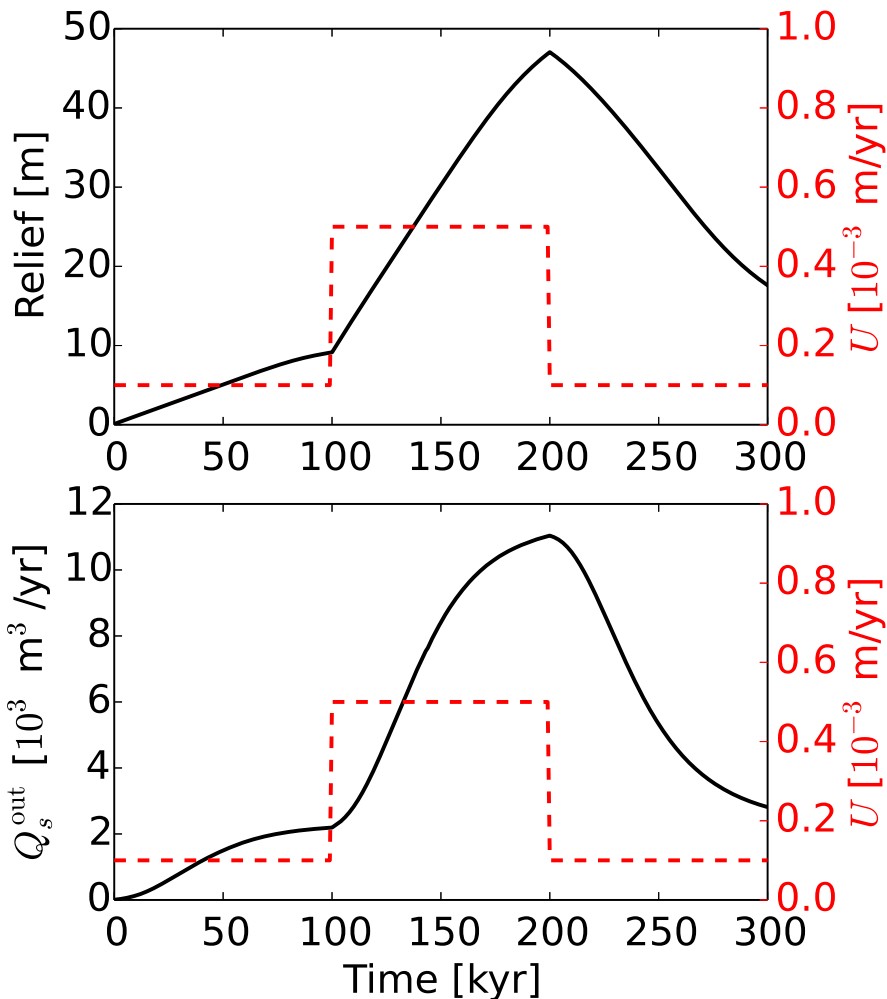

**Figure 12.** Time series of total relief in the model domain (top) and sediment flux out of the model domain ($Q_s^{\text{out}}$, bottom) shown with the imposed rock uplift rate $U$. Both relief and sediment flux increase slowly during the first 100 kyr in response to the imposed rock uplift rate of 0.0001 m/yr. By the end of the first 100 kyr, the rates of increase for both relief and sediment flux become slower, indicating that the landscape is approaching equilibrium with the rock uplift rate. Between 100 and 200 kyr, the rock uplift rate is increased by a factor of five and both relief and sediment flux increase by approximately the same factor. By the end of the 100 kyr period of high rock uplift rate, relief and sediment flux begin to equilibrate to the new, higher rock uplift rate. At 200 kyr, the rock uplift rate is reduced to its original value, and relief and sediment flux decline in response as erosion of high points on the topography outpaces rock uplift.