# Peer review of "The SPACE 1.0 model: A Landlab component for 2-D calculation of sediment transport, bedrock erosion, and landscape evolution"

_Geoscientific Model Development, 2017_

## Referee Comment (RC1) · JM Turowski (Referee) · 30 Aug 2017

In the paper, the authors develop a new formulation for river sediment transport and erosion, with a formulation that both honors conservation of mass along the stream, as in transport-limited formulations, and calculates the erosion of bedrock, as in detachment-limited formulations. While I do not have a problem with the model development and the description of the numerical implementation, the literature overview is incomplete and the introduction, review, discussion and conclusions will need to be adjusted. In particular, the author have overlooked a number of contributions (more than half of those that exist by my count, and there are not that many!) that attempted to

solve the same problem. These include the Exner-equation-based approach by Inoue et al. (JGR, 2014), the adaption of erosion-deposition models for partially alluviated beds by Turowski (WRR, 2009) and Turowski and Hodge (ESurf, 2017), the 2-D models by Nelson and Seminara (GRL, 2011 and GRL, 2012) and Inoue et al. (JHE, 2016 and ESPL, 2017), and the formulations based on St.-Venant equations (the most relevant paper here is by Fowler et al., SIAM J. Appl. Math., 2007). There might be other papers and the authors should look out for them. Really, the number of publications is not that large, and a review should encompass the entirety of the literature. I think the formulation proposed here is sufficiently different to previous models to warrant publications, but it is definitely necessary to put it into proper context. The discussion could contrast the different model formulations and highlight the differences, advantages and disadvantages of the new formulation. Finally, it would be useful to develop testable hypotheses that can be used to discriminate the various models.

Comments by line

2.20 I am not too happy with the term 'hybrid' here. This implies that two rather different approaches are put together. I rather see the two model families that are commonly termed detachment- and transport-limited as rather extreme approximations of a single approach. See also the comment to 2.30.

2.20 There are a number of important contributions missing in this overview. Inoue et al. (JGR, 2014) described a 1-D model based on an adapted Exner equation. There is also the surface-roughness model by Johnson (JGR, 2014; cited elsewhere). Turowski and Hodge (ESurf, 2017) and Turowski (WRR, 2009) adapted the erosion-deposition framework to partially alluviated beds, the latter in a stochastic context (although these papers are more concerned with cover dynamics on the reach scale, rather than sediment routing on the catchment scale). Nelson and Seminara (GRL, 2011 and 2012) and Inoue et al. (JHE, 2016, and ESPL 2017) described fully coupled 2-D models. I'd also like to point out the family of landscape evolution models that sprang from Smith and Bretherton's (WRR, 1972) seminal work. These have since been continuously

developed and expanded. Versions of these models including bedrock erosion terms have been discussed by Fowler et al. (SIAM J. Appl. Math., 2007), Smith (JGR, 2010), and Cattan et al. (Math. Geosci., 2017). The Fowler et al. paper is the most relevant here.

2.30 The work of Hodge et al. (JGR, 2012), Chatanantavet and Parker (WRR, 2008) and Turowski and Hodge (ESurf, 2017) should probably be cited here.

2.30 Here, the different concepts of sediment transport and bedrock incision models seem to be muddled. An incision law attempts to predict the bedrock erosion rate, given sediment flux, hydrodynamics, etc. A sediment transport model predicts the sediment transport rates, given the hydrodynamics. Many of the cited erosion models (such as the saltation-abrasion model or the stream power model) were not constructed to include the prediction of sediment transport rates. The assumption that the river is always under capacity, allowing to neglect mass conservation, is separate from this. In essence, there is a description of mass conservation (such as the Exner equation or the erosion-deposition framework) and a description of erosion mechanics (such as the saltation-abrasion model or the stream power model). As the authors are aware, one of these is often neglected in landscape evolution modelling – the mass conservation in the so-called detachment-limited models and the erosion mechanics in the so-called transport-limited models. The authors do seem to be aware of this distinction, as they advocate their formulation as one that might work with different erosion models.

3.2 Earth capitalized.

3.4 There have been several other potential solutions. See comment to 2.20.

3.5 Erosion-deposition models are NOT equivalent to 'under-capacity' models.

3.13 If I remember correctly, this validation is for alluvial rivers, right?

3.18 There are two papers that have done these modifications, at least partially: Turowski (WRR, 2009) extended a stochastic Markov-chain model of bedload transport to

partially alluviated beds and Turowski and Hodge (ESurf, 2017) described a 1-D model. Both these papers focus on cover dynamics rather than sediment routing.

6.21 The exponential model is functionally equivalent to that derived by Turowski et al. (2007, cited elsewhere). If H is the average height of the sediment, then this H scales with the total mass of sediment residing on the bed.

8.33 To me, 'shown' seems to be an overstatement here. Also, the meaning of state function may be unclear to readers in the current context.

9.5 It is unclear why this approach is deemed necessary and why this particular function is chosen. The motivation for a different approach seems sufficiently clear, but the authors could better describe their train of thoughts for arriving at eq. (9).

10.6 The formulation seems a bit cynical here – either the model is a good representation of reality, and then one should just have to deal with sharp discontinuities, or it is not. To choose a particular model set up to ease the analysis of the results (or to recommend it) is rather unscientific.

10.12 Eq. (11) holds only if the sediment and the water move at the same speed. The (mass) concentration is defined as M_sediment/M_water for a control volume. The mass is related to the mass transport capacity (Q_s with units kg/s) including sediment velocity V_sediment as M_sediment * V_sediment = Q_sediment * transport_length (and a similar equation for water).

11.8 Eq. (18) may be clearer if the common factors in the two terms are taken out of the parenthesis. In effect, this is a standard stream power model long profile, with an erodibility coefficient that is an inverse sum of the coefficients for bedrock and alluvium.

14.1 This sentence is rather awkward. Consider reformulating.

17.4 Earth capitalized.

20.5 Eq. (43) may be clearer if the common factors in the two terms are taken out of

the parenthesis. In effect, this is a standard stream power model long profile, with a modulating factor depending on runoff and settling velocity. It would also be interesting to quantify this factor to see how far typical values are different from one.

24.20 The test results do not indicate that the model is useful for natural settings as claimed here. They just demonstrate that the numerical implementation is working.

References

Cattan, D., and Birnir, B.: Numerical analysis of fluvial landscapes, Math. Geosci., https://doi.org/10.1007/s11004-017-9698-6, 2017.

Chatanantavet, P., and Parker, G.: Experimental study of bedrock channel alluviation under varied sediment supply and hydraulic conditions, Water Resour. Res., 44, W12446, https://doi.org/10.1029/2007WR006581, 2008.

Fowler, A. C., Kopteva, N., and Oakley, T. B.: The formation of river channels, SIAM J. Appl. Math., 67(4), 1016-1040, https://doi.org/10.1137/050629264, 2007.

Hodge, R. A., and Hoey, T. B.: Upscaling from grain-scale processes to alluviation in bedrock channels using a cellular automaton model, J. Geophys. Res. Earth Surf., 117, F01017, https://doi.org/10.1029/2011JF002145, 2012.

Inoue, T., Izumi, N., Shimizu, Y., and Parker, G.: Interaction among alluvial cover, bed roughness, and incision rate in purely bedrock and alluvial-bedrock channel, J. Geophys. Res., 119, 2123–2146, https://doi.org/10.1002/2014JF003133, 2014.

Inoue, T., Iwasaki, T., Parker, G., Shimizu, Y., Izumi, N., Stark, C. P., and Funaki, J.: Numerical simulation of effects of sediment supply on bedrock channel morphology, J. Hydr. Eng., 142, 04016014, https://doi.org/10.1061/(ASCE)HY.1943-7900.0001124, 2016.

Inoue, T., Parker, G., and Stark, C. P.: Morphodynamics of a bedrock-alluvial meander bend that incises as it migrates outward: approximate solution of permanent form,

Earth Surf. Process. Landforms, in press, https://doi.org/10.1002/esp.4094, 2017.

Nelson, P. A. and Seminara, G.: Modeling the evolution of bedrock channel shape with erosion from saltating bed load, Geophys. Res. Lett., 38, L17406, https://doi.org/10.1029/2011GL048628, 2011.

Nelson, P. A. and Seminara, G.: A theoretical framework for the morphodynamics of bedrock channels, Geophys. Res. Lett., 39, L06408, https://doi.org/10.1029/2011GL050806, 2012.

Smith, T. R.: A theory for the emergence of channelized drainage, J. Geophys. Res., 115, F02023, https://doi.org/10.1029/2008JF001114, 2010.

Smith, T. R., and Bretherton, F. P.: Stability and the conservation of mass in drainage basin evolution, Water Resour. Res., 8(6), 1506-1529, 1972.

Turowski, J. M.: Stochastic modeling of the cover effect and bedrock erosion, Water Resour. Res., 45, W03422, https://doi.org/10.1029/2008WR007262, 2009.

Turowski, J. M., and Hodge, R. A.: A probabilistic framework for the cover effect in bedrock erosion, Earth Surf. Dyn., 5, 311-330, https://doi.org/10.5194/esurf-5-311-2017, 2017.

---

## Referee Comment (RC2) · F. Clubb (Referee) · 12 Sep 2017

**General comments**

This paper derives a new formulation for modelling of channel evolution incorporating simultaneous erosion of bedrock and sediment transport, set within the Landlab modelling framework. This modelling framework has the potential to be useful to many within the earth science community by tackling a gap between the two end member scenarios of detachment- and transport- limited models: although previous models between these have been proposed, the inclusion of the authors' model within the

Landlab framework allows it to be used easily by the community and for the results of the authors' study to be reproduced, as well as being applicable over large spatial and temporal scales. The manuscript is well written, clear, and the derivation of the model is well laid out. I therefore believe that the paper is suited to publication within GMD following to some corrections which I have specified below.

**Specific comments**

Abstract: previous models have been presented in the literature that include combine erosion and deposition, which the authors review in Section 2, but there is no mention in the abstract of the novelty of the authors' approach compared to these previous methods. The abstract should describe precisely why a new modelling approach is needed for this problem.

Page 1, Lines 18 - 20: sentence needs reworded.

Page 2, Line 5: '...the superiority of one model over the other in tests against real landscapes...': I think it would be useful to include in the introduction some examples of how these different models have been tested against real landscapes. A brief review of this would be useful to the reader, and set the context for providing some demonstration of the potential of using equivalent tests for the SPACE model.

Page 4, Lines 1 - 7: The authors could expand here upon what the limitations are of the erosion-deposition models of Lague (2010) and Zhang et al. (2015).

Page 4, Lines 15-17: Other models have been developed that model mixed-bedrock alluvial channels that are not mentioned here, as noted by Reviewer 1. I think it is worth mentioning here the difference between the reach-scale approach

generally taken by the development of these mixed bedrock-alluvial models compared to the whole landscape scale that is used as a framework in this paper, to demonstrate the need for and motivation behind development of the SPACE model. It would also be good to discuss the challenges behind scaling reach-scale models up to whole landscapes.

Page 5, Eq 1: Do you think that including a variable channel width would be possible in the model, or would this be too computationally expensive to do for large spatial/temporal scales? I know this is discussed in the previous section as processes which are not treated in the model, but it would be interesting to have an idea of which processes would potentially be possible to include in future developments, and what isn't due to scale issues.

Figure 2: It seems like it would be possible to include the tool effect as well as the cover effect, where $E_r$ would decrease where $H/H* = 0$ as there are no tools for effective incision into bedrock? Is there potential for this to be included as an option in the model formulation?

Page 8, Line 33: It would be useful to explain 'state function' in some more detail here.

Page 9, Line 6: What is the justification for choosing an exponential decline in erosion at the threshold value here?

Page 12, Line 27: Is there a specific value of $q/V$ above which the model predicts detachment-limited behaviour? It would be good to clarify this here.

Page 13, Line 9: add in reference to equation 11 here.

[Figure]

Page 13, Equation 29: I like the within-cell calculation of sediment flux!

Page 14, Line 1: If $Q_s = Q_S{}^{out}$, does this imply that all of the deposition happens at the downstream node of each model cell?

Figure 4: It would be useful to have a plot of a run where you vary the ratio of $E_s$ to $D_s$ to demonstrate how this could affect the sediment thickness as well as the plots where you vary the parameters independently.

Page 18, Line 9: Have you assessed the stability of the model to the timestep? How stable is the model at greater timesteps? 1 year seems like a very short timestep if you wanted to run the model over geological timescales.

Section 6: Although testing of the model against the analytical solutions is useful in showing that the numerical implementation is working (and it's really nice to see that it can replicate both detachment-limited and transport-limited behaviour), this does not evaluate the applicability of the method to real landscapes, especially as the analytical solutions are from the same framework as that of SPACE model (e.g. detachment-limited stream power and transport-limited eqns). What is really interesting is to know how we could validate the model predictions against real landscapes. I think the paper would be improved if the authors could provide some recommendations of how their model can be tested on real landscapes (either natural or experimental), maybe as a section in the discussion (I'm aware it's a model description paper, so actually performing these validations is probably beyond the scope here).

Figure 8: This figure is really interesting. I wonder if it would be possible to compare the different concavities of the channel profiles predicted from the bedrock surface and that of the alluvial layer. Does the concavity of the profile through time give some indication of how detachment-limited or transport-limited the model is at

that point? I wonder if in real landscapes this could give an indication of transience, or for the transition between detachment- and transport-limited conditions downstream along channels.

Code availability: I like the iPython notebook, it was really easy to use and gives a good idea of the model structure.

---

## Author Comment (AC1) · 18 Oct 2017

**Response to review by F. Clubb**

Author's note: reviewer comments are reproduced here, and our responses are *italicized*.

**General comments**

This paper derives a new formulation for modelling of channel evolution incorporating simultaneous erosion of bedrock and sediment transport, set within the Landlab modelling framework. This modelling framework has the potential to be useful to many within the earth science community by tackling a gap between the two end member scenarios of detachment- and transport- limited models: although previous models between these have been proposed, the inclusion of the authors' model within the Landlab framework allows it to be used easily by the community and for the results of the authors' study to be reproduced, as well as being applicable over large spatial and temporal scales. The manuscript is well written, clear, and the derivation of the model is well laid out. I therefore believe that the paper is suited to publication within GMD following to some corrections which I have specified below.

*We thank the reviewer for taking the time to review this manuscript, and for their helpful suggestions. We have endeavored to thoroughly address all comments below.*

**Specific comments**

Abstract: previous models have been presented in the literature that include combine erosion and deposition, which the authors review in Section 2, but there is no mention in the abstract of the novelty of the authors' approach compared to these previous methods. The abstract should describe precisely why a new modelling approach is needed for this problem.

*In our revisions we have tried to stress that the novelty of our approach involves the ability to transition between transport- and detachment-limited states (which some existing models can do), the ability to erode sediment and bedrock at the same time (which some existing models can do), but most importantly the ability to do both of those in a*

[Figure]

*2-D landscape evolution model that is designed to be coupled to other models and run over landscape evolution space and time scales. Basically, we feel that we are incorporating the best features of existing models (with a slightly different mathematical formulation) into an extensible and efficient landscape evolution modeling tool. Further, we are expanding upon existing models by applying the explicit erosion/deposition framework to a mixed bedrock-alluvial channel model. We have changed the abstract to add: "Modeling landscape evolution over large spatial and temporal scales requires a model that can 1) transition freely between transport-limited and detachment-limited behavior, 2) simultaneously treat sediment transport and bedrock erosion, and 3) run in 2-D over large grids and be coupled with other surface process models.", and "SPACE improves on previous models of bedrock-alluvial rivers by explicitly calculating sediment erosion and deposition rather than relying on a flux divergence (Exner) approach."*

Page 1, Lines 18 - 20: sentence needs reworded.

*We broke the first sentence up into two to improve readability and clarity. It now reads: "Rivers are the primary agents of land-surface lowering in non-glaciated landscapes (e.g., Whipple, 2004). Erosion and sediment transport in rivers affect human river management (e.g., Graf et al., 2010), landscape mass balance (e.g., Armitage et al., 2011), and global biogeochemical cycling (e.g., Hilton, 2017)."*

Page 2, Line 5: '...the superiority of one model over the other in tests against real landscapes...': I think it would be useful to include in the introduction some examples of how these different models have been tested against real landscapes. A brief review of this would be useful to the reader, and set the context for providing some demonstration of the potential of using equivalent tests for the SPACE model.

*We have written a new section (section 2) to discuss how these models have been tested against one another. This, as noted by the reviewer, will help provide context for our later discussion of how SPACE could be tested in the field (section 9).*

Page 4, Lines 1 - 7: The authors could expand here upon what the limitations are of the erosion-deposition models of Lague (2010) and Zhang et al. (2015).

*We have re-written the review to more fully educate readers about the pros and cons of the many different modeling approaches out there. This includes a statement about the advantages of the SPACE algorithm over those two models, which are that 1) SPACE uses explicit calculation of sediment erosion and deposition instead of an Exner-style conservation rule, and 2) SPACE is easily applied to landscape evolution modeling exercises because of its status as a Landlab component.*

Page 4, Lines 15-17: Other models have been developed that model mixed bedrock alluvial channels that are not mentioned here, as noted by Reviewer 1. I think it is worth mentioning here the difference between the reach-scale approach generally taken by the development of these mixed bedrock-alluvial models compared to the whole landscape scale that is used as a framework in this paper, to demonstrate the need for and motivation behind development of the SPACE model. It would also be good to discuss the challenges behind scaling reach-scale models up to whole landscapes.

*As we note in our response to reviewer 1, we have extensively re-written and expanded our literature review (section 3) to take these other contributions into account. The new review contains a discussion of several of these models that are focused on reach-scale cover dynamics. At that point, we discuss why some of these models are likely not to scale well for landscape evolution applications (the computational cost of solving*

*2-D flow equations, for example). We hope that this shows the reader why SPACE is an important contribution.*

Page 5, Eq 1: Do you think that including a variable channel width would be possible in the model, or would this be too computationally expensive to do for large spatial/temporal scales? I know this is discussed in the previous section as processes which are not treated in the model, but it would be interesting to have an idea of which processes would potentially be possible to include in future developments, and what isn't due to scale issues.

*This is a good question. As numerous authors have pointed out, the "default" channel width closure (i.e., width goes as the square root of discharge) is computationally efficient but not physically-based. It would indeed be possible to incorporate dynamic channel width into the model. We view the approach of Lague (2010), in which stresses are partitioned between the channel bed and banks according to a trapezoidal cross-section approximation, as one potential way of incorporating width changes. There are also more complex approaches for calculating or approximating the shear stress across the channel cross-section, which we discuss in the added text below. We have not attempted to incorporate such rules into the SPACE model, and so cannot be sure what the computational cost would be, nor how that cost would scale with grid size. Our rationale for excluding dynamic width from the SPACE model is both potential computational cost but also the substantial number of parameters already necessary to describe the coupled evolution of sediment and bedrock. Adding an empirical or semi-empirical width adjustment rule would substantially increase model complexity.*

*In order to make it clear to the reader that dynamic width could be incorporated, we have added the following in section 4.3:*

*"While we employ a simple closure for channel width in which width scales as the*

*square root of water discharge (Leopold and Maddock, 1953; Wohl and David, 2008), it may be desirable for some applications to add dynamic channel width adjustments to the model, as previous work has suggested that width trades off with slope in transient channels (e.g., Finnegan et al., 2005; Turowski et al., 2006; 2009; Wobus et al., 2006; Whittaker et al., 2007; Attal et al., 2008; Lague, 2010; Yanites and Tucker, 2010). One option for incorporating dynamic width is to calculate or approximate shear-stress distributions across channel cross-sections (Kean and Smith, 2004; Wobus et al., 2006; 2008; Turowski et al., 2009). A simpler dynamic width rule can be obtained by partitioning erosive power between the bed and banks under a trapzoidal channel assumption (Flintham and Carling, 1988) as detailed in (Lague, 2010). Different approaches have different numbers of parameters and computational costs, and further work will be necessary to elucidate which advances beyond the standard empirical width closure are tractable within the SPACE landscape evolution model framework."*

Figure 2: It seems like it would be possible to include the tool effect as well as the cover effect, where $E_r$ would decrease where $H/H_* = 0$ as there are no tools for effective incision into bedrock? Is there potential for this to be included as an option in the model formulation?

*It is indeed possible to include the tools effect, but we suggest that a more effective way to do so would be to assert that $E_r$ increases with $Q_s$ until $Q_s$ reaches transport capacity (where transport capacity is $Q_s$ when erosion equals deposition, or $E_r + E_s = D_s$). The advantage of this approach is that bedrock erosion actively responds to the amount of sediment in transport, not the amount of sediment resting on the bed. The cover effect, or reduction in $E_r$ as $H/H_*$ increases, would compete with the increase in $E_r$ as $Q_s$ increases. We do not incorporate this effect into the current version of SPACE because it complicates model comparison with analytical solutions, but it could certainly be incorporated into future versions.*

*We agree with the reviewer that it would be helpful to explain to the reader how the tools effect could be incorporated. To that end, we have added two sentences to the end of section 3.3: "The "tools effect" on bedrock erosion could be incorporated into the model by assuming that $E_r$ at a given $H/H_*$ increases with $Q_s$ until $Q_s$ reaches transport capacity (which is $Q_s$ when $E_s + E_r = D_s$). Because increases in $H/H_*$ already account for the "cover effect," the $E_r$ dependence on $Q_s$ need only be positive and not decline with increasing $Q_s$ (Sklar and Dietrich, 2001; 2004; Gasparini et al., 2006; Turowski et al., 2007)."*

Page 8, Line 33: It would be useful to explain 'state function' in some more detail here.

*In response to this comment, the following comment, and comments from the other reviewer, we have re-structured and re-worded section 4.3.1 (see response to next comment below). In doing so, we no longer use the term 'state function.'*

Page 9, Line 6: What is the justification for choosing an exponential decline in erosion at the threshold value here?

*The idea here is to account for the fact that sediment entrainment and bedrock erosion thresholds in nature cannot generally be characterized by a single value. Rather, the threshold for a given population of sediment grains or a given areal exposure of bedrock is likely to be a distribution of values. The distribution will be influenced by such variables as grain size and sorting, flow history, and sediment flux for sediment, and weathering, local mineralogy, and joint spacing and orientation for bedrock (see our revised section 4.3.1). We chose an exponential function because it accounts for a smoothed threshold while still yielding the behavior expected from a single-value threshold. For example, when stream power is far above the user-defined threshold, excess stream*

*power is $\omega - \omega_c$ just as in single-threshold cases. Conversely, when stream power is far below the defined threshold, the threshold term approximately equals $\omega$ (the available stream power), meaning that entrainment or erosion is effectively zero. The other benefit of choosing an exponential function over some other form is that it does not require any additional model parameters because the user-defined threshold scales the exponential function.*

*To make all of this clear to the reader, we have re-written section 4.3.1. We focus on summarizing support for the idea of a distribution of thresholds rather than a single number, and we have added a sentence explaining that we prefer the exponential because it adds no model parameters: "We chose an exponential function because it allows for smoothing of entrainment and erosion thresholds, and therefore honors the reality that such thresholds tend to be distributions of values rather than a single value, without adding any model parameters."*

Page 12, Line 27: Is there a specific value of q/V above which the model predicts detachment-limited behaviour? It would be good to clarify this here.

*In the erosion/deposition model presented by Davy and Lague (2009), there is a dimensionless number $\frac{V}{r}$ that predicts the behavior. When $\frac{V}{r} > 1$, behavior is transport-limited and vice versa. However, this only holds when eroding a single layer, as incorporating multiple layers increases the number of factors that influence sediment entrainment (for example, in SPACE, $H/H_*$ has a strong influence). There is not a specific value that governs the DL/TL transition in the SPACE model, because the dimensionless ratios presented in this section do not incorporate all of these effects. Still, we think that it is worth presenting the simple case from Davy and Lague (2009), so we have added text to say:*

*"Specifically, following Davy and Lague (2009), we can define a dimensionless number*

$\frac{V}{r}$ *that governs the transition between detachment-limited and transport-limited dynamics. In the sediment-only case (when $H \gg H_*$), or in the bedrock-only case ($H = 0$), $\frac{V}{r} > 1$ gives transport-limited behavior and $\frac{V}{r} < 1$ results in detachment-limited behavior (Davy and Lague, 2009). In cases where sediment and bedrock are eroded simultaneously, especially if there is a significant erodibility contrast between the two, the behavior is not so easily predicted and will generally contain contributions from both detachment and transport limitations."*

Page 13, Line 9: add in reference to equation 11 here.

*We have added the reference.*

Page 13, Equation 29: I like the within-cell calculation of sediment flux!

*Thank you!*

Page 14, Line 1: If $Q_s = Q_s^{out}$, does this imply that all of the deposition happens at the downstream node of each model cell?

*Good question. The local analytical solution for $Q_s$ was designed to account for the fact that 1) rapid sediment entrainment should decline over a cell length as it depletes the available sediment, and 2) rapid deposition should decline over a cell length as $Q_s$ gets smaller. Using $Q_s = Q_s^{out}$ allows us to achieve this outcome, whereas using $Q_s = Q_s^{in}$ would allow for over-erosion or over-deposition, depending on the value of $Q_s^{in}$.*

*To answer the reviewer's specific question: using $Q_s = Q_s^{out}$ allows us to correctly account for the sediment deposited within a cell, but that sediment is deposited over*

*the whole length of the cell.*

*In response to this comment as well as one from the other reviewer, we have slightly altered this sentence to clarify why we equate $Q_s$ and $Q_s^{out}$ for the deposition term: "Because sediment deposition in a cell depends on both $Q_s^{\text{in}}$ from upstream and sediment entrained from the cell itself, we can substitute $Q_s^{\text{out}}$ for $Q_s$ in the deposition term. Eq. (29) may then be solved to yield the local analytical solution for $Q_s$ within a model cell:"*

Figure 4: It would be useful to have a plot of a run where you vary the ratio of $E_s$ to $D_s$ to demonstrate how this could affect the sediment thickness as well as the plots where you vary the parameters independently.

*We have added two panels to Figure 4 to address this comment. One shows the results of varying $E_s$ and $D_s$ simultaneously in order to vary the ratio between the two, as requested by the reviewer. The other shows the result of keeping a constant ratio of $E_s$ to $D_s$ while the magnitudes of the two parameters change. While the reviewer did not ask for this specific addition, we felt like the behavior (that lower values of $E_s$ and $D_s$, even if their ratio is the same, result in a longer adjustment timescale for $H$) was worth pointing out.*

Page 18, Line 9: Have you assessed the stability of the model to the timestep? How stable is the model at greater timesteps? 1 year seems like a very short timestep if you wanted to run the model over geological timescales.

*The model is generally stable at timesteps greater than one year. The example in the code guide is stable at 10-year timesteps with the default parameterization, but becomes unstable at 50-year timesteps. We used a 1-year timestep in the text and*

*example code 1) to ensure high numerical accuracy for comparison with analytical solutions and 2) to allow readers/users to change parameters with a lower likelihood of making the model unstable. We have heuristically assessed model stability with respect to the timestep over different model parameterizations and grid resolutions, and found that in general, 10-year timesteps are stable in most common modeling scenarios. However, we are unaware of a Courant-Friedrichs-Levy-type criterion that could be formally defined for the numerical implementation of SPACE. The stability of SPACE depends strongly on the strength of the $E_s$ and $D_s$ terms. For example, when $K_s$ and $V$ are high such that sediment is easily entrained and deposited, the model is stable at a smaller timestep than it is when $K_s$ and $V$ are low (i.e., when the river is acting in a detachment-limited way).*

*A simple back-of-the-envelope calculation for a stability criterion is to assume a sediment-mantled bed of a given slope and calculate the time required for an upstream node to erode enough that its slope is lower than its downstream neighbor. For example, consider an end-member case of $H \gg H_*$ in which two nodes separated by distance $dx$ lie along a slope of 0.001 such that the vertical drop between them is 0.1 m. In this case, neglecting any entrainment threshold and assuming no sediment flux from upstream, $E_s = K_s q S^n$. The maximum timestep that will prevent the upstream node from becoming lower than the downstream node is 0.1 m divided by $E_s$. As such, the maximum stable timestep shrinks as $K_s$ and $q$ increase.*

*In response to the reviewer's question, we have added text to section 6 briefly making the point that the stability of SPACE is subject to its specific parameterization, but also that we have generally found it to be stable for 10-year timesteps. This new text reads: "While the SPACE 1.0 component is stable at 10-year timesteps under most conditions, we use a timestep of 1 year here to maximize numerical accuracy for comparison with analytical solutions."*

Section 6: Although testing of the model against the analytical solutions is useful in

showing that the numerical implementation is working (and it's really nice to see that it can replicate both detachment-limited and transport-limited behaviour), this does not evaluate the applicability of the method to real landscapes, especially as the analytical solutions are from the same framework as that of SPACE model (e.g. detachment-limited stream power and transport-limited eqns). What is really interesting is to know how we could validate the model predictions against real landscapes. I think the paper would be improved if the authors could provide some recommendations of how their model can be tested on real landscapes (either natural or experimental), maybe as a section in the discussion (I'm aware it's a model description paper, so actually performing these validations is probably beyond the scope here).

*This comment was echoed by the other reviewer as well, and we have added a section to the paper (section 9.1) to address how the model might be validated against real landscapes. As the reviewer notes, we view performing such tests as beyond the scope of this model description paper, but we hope that our discussion will provide avenues for future model validation efforts. In section 9.1 we discuss 1) potential validation of the steady-state predictions that SPACE makes for channel slope and sediment thickness, and 2) potential validation of the transient predictions, including the transient differences in concavity between the sediment surface and the bedrock surface, as discussed in the reviewer's next comment.*

Figure 8: This figure is really interesting. I wonder if it would be possible to compare the different concavities of the channel profiles predicted from the bedrock surface and that of the alluvial layer. Does the concavity of the profile through time give some indication of how detachment-limited or transport-limited the model is at that point? I wonder if in real landscapes this could give an indication of transience, or for the transition between detachment- and transport-limited conditions downstream along channels.

*This is an interesting point and we have endeavored to briefly address it in the new section on comparing SPACE to field data (see response to comment above). One potential pitfall that would need further exploration is the role of discharge variability. When we measure something like sediment thickness (or the surface profile of a sediment layer) in the field, it is unclear whether that particular measurement is representative of the temporal mean, given that the most recent flow events can have a strong control on sediment thickness. To fully explore the benefits/problems of the comparison the reviewer suggests is a topic for another day, but we have added some text to section 9.1 to make readers aware of the potential opportunity:*

*"In the transient case, the relationship between the longitudinal profile of the bed sediment surface and that of the bedrock may be useful for validating SPACE model predictions. For example, Fig. 8 shows that for an uplifting landscape with zero initial sediment thickness, SPACE predicts bottom-to-top alluviation of the channel profile. In this case, the sediment surface does not reflect the steepened reach commonly associated with the propagation of transient signals up a river profile, while the bedrock beneath does. The prediction of SPACE is therefore that in a transient river profile with some amount of bed sediment, the concavity of the sediment surface is not expected to match that of the bedrock surface. The difference in concavity between the sediment surface and the bedrock surface should then decline as channels approach steady state, a prediction that is testable in a landscape where channels exist in different stages of transient adjustment. It is important to remember that the sediment thickness predicted by SPACE is a spatial average within a model cell. Further, using realistic (i.e., time-varying) flow distributions to force the model would result in temporal variability in sediment thickness (Lague et al., 2005; Lague, 2010), complicating the interpretation of sediment thickness values from a specific field campaign. While testing the steady-state predictions of SPACE is likely feasible in well-constrained landscapes, the transient dynamics may be best explored in a laboratory setting."*

Code availability: I like the iPython notebook, it was really easy to use and gives a good idea of the model structure.

*Thank you!*

---

## Author Comment (AC2) · 18 Oct 2017

**Response to review by J. Turowski**

Author's note: reviewer comments are reproduced here, and our responses are *italicized*.

**General comments**

[Figure]

In the paper, the authors develop a new formulation for river sediment transport and erosion, with a formulation that both honors conservation of mass along the stream, as in transport-limited formulations, and calculates the erosion of bedrock, as in detachment-limited formulations. While I do not have a problem with the model development and the description of the numerical implementation, the literature overview is incomplete and the introduction, review, discussion and conclusions will need to be adjusted. In particular, the author have overlooked a number of contributions (more than half of those that exist by my count, and there are not that many!) that attempted to solve the same problem. These include the Exner-equation-based approach by Inoue et al. (JGR, 2014), the adaption of erosion-deposition models for partially alluviated beds by Turowski (WRR, 2009) and Turowski and Hodge (ESurf, 2017), the 2-D models by Nelson and Seminara (GRL, 2011 and GRL, 2012) and Inoue et al. (JHE, 2016 and ESPL, 2017), and the formulations based on St.-Venant equations (the most relevant paper here is by Fowler et al., SIAM J. Appl. Math., 2007). There might be other papers and the authors should look out for them. Really, the number of publications is not that large, and a review should encompass the entirety of the literature. I think the formulation proposed here is sufficiently different to previous models to warrant publications, but it is definitely necessary to put it into proper context. The discussion could contrast the different model formulations and highlight the differences, advantages and disadvantages of the new formulation. Finally, it would be useful to develop testable hypotheses that can be used to discriminate the various models.

*We thank the reviewer for taking the time to review our manuscript, and for their helpful comments. In response to the reviewer's point that several items are missing from our literature review, we have re-written and expanded the introduction (section 1) and literature review section (section 3). The review now includes almost all of the contributions noted by the reviewer. We have also added a section (section 9.3) dedicated to elucidating the similarities and differences between our model and previous models. In response to the reviewer's comment noting that testable hypotheses would be useful*

*(which was echoed by the other reviewer), we have added section 9.1, which discusses some of the testable hypotheses generated by the examples we present in the paper. Below, we address the reviewer's specific line-by-line comments.*

**Specific comments**

2.20 I am not too happy with the term 'hybrid' here. This implies that two rather different approaches are put together. I rather see the two model families that are commonly termed detachment- and transport-limited as rather extreme approximations of a single approach. See also the comment to 2.30.

*We had been using the term 'hybrid' following Hobley et al (2011), but we have removed that term from the paper. At the end of the introduction, we introduce our review by saying that we will "review existing models that fall between these two end-members."*

2.20 There are a number of important contributions missing in this overview. Inoue et al. (JGR, 2014) described a 1-D model based on an adapted Exner equation. There is also the surface-roughness model by Johnson (JGR, 2014; cited elsewhere). Turowski and Hodge (ESurf, 2017) and Turowski (WRR, 2009) adapted the erosion-deposition framework to partially alluviated beds, the latter in a stochastic context (although these papers are more concerned with cover dynamics on the reach scale, rather than sediment routing on the catchment scale). Nelson and Seminara (GRL, 2011 and 2012) and Inoue et al. (JHE, 2016, and ESPL 2017) described fully coupled 2-D models. I'd also like to point out the family of landscape evolution models that sprang from Smith and Bretherton's (WRR, 1972) seminal work. These have since been continuously developed and expanded. Versions of these models including bedrock erosion terms have been discussed by Fowler et al. (SIAM J. Appl. Math., 2007), Smith (JGR, 2010),

and Cattan et al. (Math. Geosci., 2017). The Fowler et al. paper is the most relevant here.

*We have restructured the review to incorporate these contributions (the only ones we do not include are the Smith (2010) and Cattan et al (2017) papers. We felt that the focus of Smith (2010) on subcritical vs supercritical flow, and the focus of Cattan et al (2017) on numerical techniques made them less applicable than Smith and Bretherton (1972) and Fowler (2007), which we now discuss.*

*In response to this comment and one from the other reviewer asking for a discussion of reach-scale vs. landscape-scale models, we have added a paragraph to the review discussing the differences in approach between the reach-scale cover models and landscape evolution models incorporating bedrock-alluvial dynamics.*

2.30 The work of Hodge et al. (JGR, 2012), Chatanantavet and Parker (WRR, 2008) and Turowski and Hodge (ESurf, 2017) should probably be cited here.

*We now include discussions of these three contributions in the review, as well as Chatanantavet and Parker (2009).*

2.30 Here, the different concepts of sediment transport and bedrock incision models seem to be muddled. An incision law attempts to predict the bedrock erosion rate, given sediment flux, hydrodynamics, etc. A sediment transport model predicts the sediment transport rates, given the hydrodynamics. Many of the cited erosion models (such as the saltation-abrasion model or the stream power model) were not constructed to include the prediction of sediment transport rates. The assumption that the river is always under capacity, allowing to neglect mass conservation, is separate from this. In essence, there is a description of mass conservation (such as the Exner equation or

the erosion-deposition framework) and a description of erosion mechanics (such as the saltation-abrasion model or the stream power model). As the authors are aware, one of these is often neglected in landscape evolution modelling – the mass conservation in the so-called detachment-limited models and the erosion mechanics in the so-called transport-limited models. The authors do seem to be aware of this distinction, as they advocate their formulation as one that might work with different erosion models.

*In our revised review, we have been careful to state whether each model discussed a) conserves sediment and b) contains an incision rule. As we note in comment 3.4, we have deleted the term 'under-capacity' because of the confusion around that term existing in the literature.*

3.2 Earth capitalized.

*Fixed.*

3.4 There have been several other potential solutions. See comment to 2.20.

*As discussed above, we have restructured the review to include the contributions noted by the reviewer. We feel that the revised review both better acknowledges the general body of work on this topic and better categorizes existing models than the original manuscript did.*

3.5 Erosion-deposition models are NOT equivalent to 'under-capacity' models.

*We have deleted the term 'under-capacity' due to its confusing use in the literature.*

*We included this comment originally because Davy and Lague (2009) note that their erosion-deposition model, when the sediment transport length scale is constant in space, becomes equivalent to the model of Beaumont et al (1992). However, we now understand that the term 'under-capacity' in that context may be confused with the detachment-limited assumption (e.g., mass conservation may be neglected). As such, we do not use the term in the revised paper.*

3.13 If I remember correctly, this validation is for alluvial rivers, right?

*That is correct. To make this clear, we have changed this sentence to read: "The erosion-deposition framework, validated for alluvial rivers by the laboratory experiments of. . ."*

3.18 There are two papers that have done these modifications, at least partially: Turowski (WRR, 2009) extended a stochastic Markov-chain model of bedload transport to partially alluviated beds and Turowski and Hodge (ESurf, 2017) described a 1-D model. Both these papers focus on cover dynamics rather than sediment routing.

*As discussed above, we have re-written the review to incorporate these two (and other) contributions. Additionally, we have discussed in the review the differences between approaches that focus on cover dynamics vs. those that focus on catchment-scale sediment routing and landscape evolution.*

6.21 The exponential model is functionally equivalent to that derived by Turowski et al. (2007, cited elsewhere). If H is the average height of the sediment, then this H scales with the total mass of sediment residing on the bed.

*We have referenced Turowski et al. (2007) in this sentence.*

8.33 To me, 'shown' seems to be an overstatement here. Also, the meaning of state function may be unclear to readers in the current context.

*We have substantially revised section 4.3.1 to make our meaning more clear. In so doing, we have replaced "shown" with ". . . the sediment transport literature suggests. . . ". We have also removed the term "state function."*

9.5 It is unclear why this approach is deemed necessary and why this particular function is chosen. The motivation for a different approach seems sufficiently clear, but the authors could better describe their train of thoughts for arriving at eq. (9).

*In our re-writing of section 4.3.1, we have clarified our thought process with regards to the development of the smoothed-threshold functions. The added text is copied below.*

*"If a distribution of thresholds exists, erosion should decline to zero not exactly when available stream power drops below the user-defined threshold as would be the case for a standard threshold model, but when available stream power is significantly below the defined threshold. As available stream power becomes larger than the defined threshold, entrainment and erosion should increase smoothly as a greater portion of the distribution of thresholds is exceeded. In the limit where available stream power is many times greater than the user-defined threshold, available stream power should simply be reduced by the user-defined threshold. An exponential function describing the increase in entrainment/erosion as available stream power increases relative to threshold stream power satisfies these requirements without adding any model parameters. We include an optional exponential expression for threshold stream power. . . "*

10.6 The formulation seems a bit cynical here – either the model is a good represen-
tation of reality, and then one should just have to deal with sharp discontinuities, or it
is not. To choose a particular model set up to ease the analysis of the results (or to
recommend it) is rather unscientific.

*We have re-focused section 4.3.1 and deleted the sentences about the relative ease
of model analysis. Our main purpose with the smoothed-threshold approach is to pro-
vide a mathematically simple way to honor the reality that entrainment and erosion
thresholds are not generally simple constants, and to do so without adding any addi-
tional model parameters. We believe that exponential smoothing of the erosion and
entrainment thresholds is a reasonable representation of reality. To avoid confusing
the reader and/or distracting from this main point, we deleted the sentences relating to
how discontinuities in a model response surface can hinder analysis.*

10.12 Eq. (11) holds only if the sediment and the water move at the same speed. The
(mass) concentration is defined as $M_{sediment}/M_{water}$ for a control volume. The mass is
related to the mass transport capacity ($Q_s$ with units kg/s) including sediment velocity
$V_{sediment}$ as $M_{sediment} * V_{sediment} = Q_{sediment} *$ transport_length (and a similar equation
for water).

*We have added the following sentence to the end of section 4.4: "Eq. (11) assumes
that sediment and water move at the same speed such that all changes in $\frac{Q_s}{Q}$ are driven
by erosion and deposition."*

11.8 Eq. (18) may be clearer if the common factors in the two terms are taken out of
the parenthesis. In effect, this is a standard stream power model long profile, with an
erodibility coefficient that is an inverse sum of the coefficients for bedrock and alluvium.

*We have added equations making this change. The form of equation 18 was intended to facilitate comparison with Davy and Lague (2009), who showed their slope-area relationships in a similar way. However, we agree with the reviewer's point that putting the S-A relationship in a form more familiar to the basic stream power prediction will likely be helpful for readers. We have added one more step to our derivation (now equation 19) in which we separate $U$ and $A^m$ from the two additive terms. In addition, we take this opportunity to add text to address the reviewer's point (made in comment on 20.5) that constraining $\frac{V}{r}$ in natural channels would be useful. Our additional text and equations are shown below.*

"*Eq. (18) may be rearranged to show that SPACE predicts a standard stream power slope-area relationship modulated by $\frac{V}{r}$ as well as sediment and bedrock erodibility:*"

$$S = \left[ \frac{V}{K_s r} + \frac{1}{K_r} \right]^{1/n} U^{1/n} A^{-m/n}. \tag{1}$$

"*The ratio between the effective settling velocity $V$ and the runoff rate $r$ controls the relative importance of the bedrock and alluvial components of the steady-state channel slope. In the simplified case of $K_s = K_r$, a ratio of $\frac{V}{r} = 1$ would indicate equal contributions from the two regimes. Quantifying $\frac{V}{r}$ for natural systems could therefore give a valuable indication of process dynamics in natural channels.*"

14.1 This sentence is rather awkward. Consider reformulating.

*We have changed this sentence to read: "Because sediment deposition in a cell depends on both $Q_s^{\text{in}}$ from upstream and sediment entrained from the cell itself, we can substitute $Q_s^{\text{out}}$ for $Q_s$ in the deposition term. Eq. (29) may then be solved to yield the local analytical solution for $Q_s$ within a model cell:"*

17.4 Earth capitalized.

*Fixed.*

20.5 Eq. (43) may be clearer if the common factors in the two terms are taken out of the parenthesis. In effect, this is a standard stream power model long profile, with a modulating factor depending on runoff and settling velocity. It would also be interesting to quantify this factor to see how far typical values are different from one.

*We have added one last step in our derivation (now equation 45) to address this point. The new equation, shown below, should be more familiar to people used to working with the standard stream power model.*

*"Eq. (45) may be re-written to show that it predicts a standard stream power slope-area relationship that is modified by the ratio of settling velocity to effective runoff:"*

$$S = \left[\frac{V}{r} + 1\right]^{1/n} \left[\frac{U}{K_s}\right]^{1/n} A^{-m/n}. \tag{2}$$

*In response to the comment about quantifying $V/r$, we agree that because this ratio is an important control on the transport-limited vs. detachment-limited model behavior, it is worth quantifying. At this point however, we feel that the constraints on $V$ are too poor to facilitate adequate comparison. We intend to focus future work on unpacking $V$ so it no longer lumps together so many processes and variables. We have added text in section 4.5 (after equation 19) to emphasize the importance of this ratio to the reader, and to suggest that quantifying it in natural channels could be a valuable exercise.*

*"The ratio between the effective settling velocity $V$ and the runoff rate $r$ controls the relative importance of the bedrock and alluvial components of the steady-state channel slope. In the simplified case of $K_s = K_r$, a ratio of $\frac{V}{r} = 1$ would indicate equal*

*contributions from the two regimes. Quantifying $\frac{V}{r}$ for natural systems could therefore give a valuable indication of process dynamics in natural channels."*

24.20 The test results do not indicate that the model is useful for natural settings as claimed here. They just demonstrate that the numerical implementation is working.

*We have removed this sentence, as the preceding sentence already makes the point that the numerical implementation is working.*

———————————————————

---

## Author Response (AR2)

November 9, 2017

Dear Dr. Neal,

Thank you for considering our revised manuscript. Please find below our reply to Dr. Turowski with our comments italicized. We are also attaching a tracked-changes PDF, with deletions struck through and additions italicized. The only significant change to the manuscript is that we have restructured the literature review to align with the suggestions by Dr. Turowski, without substantially changing the substance or scope of the review. Specifically, this involves putting the reviewed models into classifications based on whether they describe sediment mass conservation, bedrock erosion, or both. We have made all of the minor corrections suggested by the reviewer, including fixing the discharge-area relation to ensure that the equation is dimensionally consistent. We feel that the revised manuscript is greatly improved, and that it will be of interest to the readers of *GMD*.

Thank you for considering our revised manuscript,

Charles M. Shobe, on behalf of all coauthors

**RESPONSE TO REVIEW BY J. TUROWSKI**

CHARLES M. SHOBE, ON BEHALF OF ALL COAUTHORS

Author's note: reviewer comments are reproduced here, and our responses are *italicized*.

**General comments**

Dear authors, thanks for the revisions on the paper. I have a large number of minor comments. I found a few equations that are not dimensionally balanced (e.g., $q = A^m$), so please check that everything is correct. All the best, Jens

*We thank the reviewer for taking the time to review our revised manuscript. The reviewer's most significant comments concern 1) the organization of the literature review and 2) dimensionally balancing the area-discharge relation. We have re-structured the review to align with the reviewer's suggestion, and altered the relevant equations to ensure that dimensions balance. Additionally, we have addressed each of the reviewer's specific comments below.*

**Specific comments**

1.13 Earth capitalized.

*Fixed.*

2.2 I would rather say that river incision has commonly been modelled with two end members.

*Fixed. The sentence now reads: "River incision has commonly been modeled with one of two end members:..."*

2.25 and following. The review is quite good, but I am not sure I agree with the outlook. As I already stated in the last round of reviews, for a minimum physical description of the landscape, we need two components: an equation describing mass conservation and an equation describing sediment production (i.e., an erosion law). When I read the early literature, I get the impression that the authors who constructed detachment-limited and transport-limited models in the first place were quite aware that these are end-member approximations of this minimum

description (with transport-limited models neglecting sediment production and detachment-limited models neglecting mass conservation). In this context, it seems not to be sensible to expect that the one or the other model can describe any landscape. Rather, in field studies, the pertinent question is in what kind of landscapes the one or the other approximation can be applied. In this outlook, the outcomes of Valla et al. and Hobley et al. are not contradictory. Different landscapes obviously lead to different outcomes. The need for the inclusion of both fundamental equations (mass conservation and sediment production) arises from the process observations that availability of alluvial sediment strongly influences erosion rates, for example through the tools and cover effects. This insight forbids neglecting mass conservation when describing bedrock rivers.

*We have revised section 2 to highlight the points made by the reviewer. We kept the example of Valla's and Hobley's work, but no longer state that the results are contradictory, and instead use their work as an example of how the different end-member models are not applicable to all landscapes. We end section 2 by turning to the development of models that include both mass conservation and sediment production/bed erosion.*

3.12 / section 3: This paragraph is helpful, but terminology and concepts are not currently clearly exposed. I do not agree with the hard boundaries the authors propose here for the four classes. For example, the model described in my recent paper with Rebecca Hodge in ESurf is a description of mass conservation within a channel with a partially alluviated bed. Although we have not included erosion in that paper, it would be straight-forward to do. A similar statement could be made for the Exner-based framework by Inoue et al. The classification put forward here seems to be more aligned with the aims of the particular papers in which they were proposed, and seems therefore artificial, rather than giving a classification that springs from the description of the physics. My suggestion would be to discriminate descriptions of mass conservations (e.g., Inoue et al., Turowski and Hodge, Zhang et al., but also Nelson and Seminara) and descriptions of bedrock erosion (e.g., saltasion-abrasion model and derivatives, stream power etc), and finally combinations of these (Nelson and Seminara, Davy and Lague, the present paper).

*We have reorganized the literature review to conform to the reviewer's suggestions. We now first discuss descriptions of mass conservation, then descriptions of bed erosion, then models that combine both. We have left in place the reach-scale vs. landscape-scale paragraph that was explicitly requested by our other reviewer.*

3.14 In my view, these are not detachment-limited models. Sediment-flux-dependent models come out of process observations and they aim to quantify

the sediment-production equation. They can be used within a detachment-limited framework, but they don't have to be used as such. An example is the saltation-abrasion model: the mechanistic model was described in the Sklar & Dietrich 2004 paper, the model implications for channel long profiles in a detachment-limited framework were explored in the Sklar & Dietrich 2006 paper and the recent Zhang et al. 2014 paper uses the model in a framework that includes mass conservation.

*This is a good point. In the revised literature review we have avoided describing sediment-flux-dependent bedrock erosion models as detachment-limited, and instead we simply classify them as one type of model for bedrock erosion/sediment production.*

3.29 I suggest to delete 'Depending on the specific model formulation used' – tools and cover effect may be active independent of specific model formulations. There are just some models that include a description of these effects and others do not (whether a model of the one or of the other category is a good description of reality is a different matter?)

*We have deleted the suggested clause. The sentence now begins: "Sediment may act as…"*

3.31 review by Hobley et al.

*Fixed.*

6.16 similar to what?

*We have changed this sentence to clarify our meaning, which is that the sediment transport and bedrock erosion formulations of Inoue et al. (2016, 2017) are similar to those of Inoue (2014), but have been expanded to 2-D. The sentence now reads: "Inoue (2016) and Inoue (2017) expanded the sediment conservation and bedrock erosion formulations of Inoue et al. (2014) to 2-D to allow spatial variability in alluvial thickness and bedrock erosion along both the channel length and width."*

7.22 I suggest to remove this sentence and let the community judge the significance of the contribution.

*We have softened the language in this sentence such that we no longer describe the model as a "significant advance." However, the other reviewer had requested explicit clarification of the advances of the SPACE model over previous work, so we would prefer to leave this sentence in. The re-worded sentence now reads: "Our new model advances beyond sediment-flux-dependent bedrock incision models…"*

7.34 ...used by Sklar and Dietrich...

*Fixed.*

8.21 I suggest to use the term entrainment for alluvium, to prevent confusion with bedrock erosion

*We agree and have made this change.*

9.19 ...discussed by ...

*Fixed.*

10.5 the equation $q = A^m$ needs a multiplicative coefficient; otherwise it is not dimensionally balanced.

*Fixed. We have re-written this equation as $q = k_q A^m$, where $k_q$ is an empirical constant that is subsumed into the erodibility constants for sediment and bedrock. The sentence now reads: "The simplest is $q = k_q A^m$ where $A$ is drainage area, $m$ is a scaling exponent (generally $\approx 0.5$) designed to reflect downstream width changes and discharge–area scaling (Leopold and Maddock, 1953; Snyder et al., 2003; Wohl and David, 2008), and $k_q$ is an empirical constant, with units dependent on the value of $m$, that is subsumed into $K_s$ and the rock erodibility parameter $K_r$."*

10.6 to my knowledge, the scaling exponent is typically 0.7 when bankfull discharge is used and 1 when the long-term average discharge is used. The latter can be argued from mass conservation. See Snyder et al. 2003 for a brief discussion and literature overview.

*We understand the reference the reviewer is making, but here we demonstrate that the scaling exponent should be $\approx 0.5$ as in our manuscript because we are incorporating both a discharge-area relationship and a width-discharge relationship, not simply the discharge-area relationship to which the reviewer is referring.*

*Our understanding is that Snyder (and others) suggest a scaling between volumetric water discharge and drainage area of $Q = k_q A^c$ where, as the reviewer notes, $c$ can vary from 0.7 to 1 depending on the hydrology. However in the manuscript we are defining a relation between discharge per unit width ($q$) and drainage area, which means that our relationship must also include the scaling between volumetric water discharge and channel width. Following Snyder et al (2003), $q = Q/w$ where $Q = k_q A^c$ and $w = k_w Q^b$. Combining these three relationships, we get $q = \frac{k_q A^c}{k_w k_q^b A^{bc}}$, which simplifies to $q \approx A^{c-b}$, plus the collection of constants. Snyder et al (2003) suggest that $c \approx 1$ both in theory for average discharge (as noted by the*

*reviewer) and in their calibrated cases, and they cite Leopold and Maddock (1953) and Montgomery and Gran (2001) in reporting values of $b \approx 0.5$.*

*We have not changed our description of the scaling exponent $m$, because our chosen approximate value of $0.5$ is supported by theory. However, to avoid other readers having this confusion, we have altered the relevant sentence to say "...$m$ is a scaling exponent (generally $\approx 0.5$) designed to reflect downstream width changes and discharge–area scaling." We have also added a citation to Snyder et al (2003).*

11.10 by Lague

*Fixed.*

15.5 should there be a factor accounting for the bedload fraction?

*It is correct that when the fraction of fine sediment $F_f$ is not negligible, $(1 - F_f)$ would be included in this equation. However, we stated at the beginning of section 4.5 that "we assume for the purposes of this derivation that...$F_f$ and $\phi$ are both negligible." As such, we feel that it would be confusing to include $(1 - F_f)$ in this equation and the subsequent steps of the derivation. In our opinion, it is preferable to preserve the simplicity of the derivation, and the interested reader will be able to add the extra complexity associated with erosion thresholds, $F_f$, and $\phi$.*

15.12 Again, $q = A^m$ is not dimensionally balanced.

*Fixed (see response to comment 10.5).*

27.27 Earth capitalized.

*Fixed.*

27.29 Can you give the appropriate references here?

*We have added references to Lague (2010), Nelson and Seminara (2011), Inoue et al (2014, 2016, 2017), and Zhang et al (2015). We have also removed "2-D" from this sentence to prevent readers being confused by its two potential definitions (e.g., 2-D modeling of a river bed or application of a model over a 2-D landscape). We have amended this sentence to say that models that "act over 2-D landscapes" are uncommon.*

29.25 I guess the citation here should be Turowski et al., 2009.

*Fixed.*

29.34 Even in plucking-dominated environments, tools may play a large role by driving crack extension. This process is known as macro-abrasion.

*We have altered this sentence to include the possibility of macroabrasion: "However, there is substantial field evidence indicating that mobilized bedload can be an important erosive agent, both by detaching bedrock particles and extending fractures through macroabrasion."*

30.1 . . . is underpredicted. . .

*Fixed.*

30.21 . . . as does the model by Nelson and Seminara

*Fixed.*

32.1 'necessary' used twice in this sentence

*We have deleted the second instance.*

32.9 Earth capitalized.

*Fixed.*

32.11 'unique'?

*We have removed 'unique' from this sentence.*

[revised manuscript text omitted]